# Exploring Social Sustainability Handprint—Part 1: Handprint and Life Cycle Thinking and Approaches

Roope Husgafvel 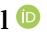

Department of Bioproducts and Biosystems, Aalto University, FI-00076 Espoo, Finland; roope.husgafvel@aalto.fi

**Abstract:** Sustainable development and sustainability encompass a strong focus on the advancement of sustainable societies, social sustainability, and overall well-being of people both now and in the future. These goals also highlight sustainable social/society–environment relationships and interfaces to promote sustainable development of both people and the planet. The promotion of social sustainability requires leadership, management, and assessment by organizations and people. This study explored social sustainability handprints from the perspective of handprint and life cycle thinking and approaches using qualitative research approaches. It addressed a clear gap in research and aimed at exploring, discovering, analyzing and synthetizing the main implications of these frameworks for the creation and assessment of the social sustainability handprint development. It was recognized that there are multiple ways to create social sustainability handprints, such as positive changes, actions, innovations, and impacts. The same applies to assessments that can be based on, for example, handprint and life cycle thinking and approaches, sustainability management, assessment and indicators, and sustainability science. The findings highlight the broadness and diversity of approaches, opportunities, and possibilities related to both the creation and assessment of social sustainability handprints. Additionally, they suggest that particular focus is needed, for example, on comprehensive approaches that take into account specific contexts, locations, cultures, scales, conditions, characteristics, perspectives, and stakeholders.

**Keywords:** social sustainability handprint; handprint; life cycle; thinking; approach; sustainability; sustainable development

## 1. Introduction

The handprint concept and approach was presented by the Centre for Environment Education (CEE) in 2007 at UNESCO's 4th International Conference on Environmental Education. The themes of this event that provide context for the handprint evolution, encompassing, e.g., (1) the principles of sustainability; (2) the contribution of work and lifestyles to the well-being of all life; (3) human rights, social justice, and gender equality; and (4) the need for human lifestyles to support ecological integrity and the climate crisis [1]. In this context, the handprint related focus areas and approaches included, e.g., action towards sustainability, education for sustainable development, positive action towards biodiversity conservation, and focus on collective and individual actions to solve environmental problems [2].

The handprint concept and approach can be very useful for modern and forward-thinking organizations, societies, and people, as it provides a good overall basis for the promotion, management, and assessment of social sustainability, including social/society–environment relationships and interfaces. The handprint approach can support addressing global challenges and promote innovation and collaboration among multiple actors, including the creation of ripple effects of positive impacts for all actors that want to promote sustainable development [3]. In addition, handprints are promoting innovation in an interconnected world, and they promote a systemic thinking approach to sustainability instead of focusing only on a linear thinking perspective (e.g., footprints) of sustainability

management and assessment [4]. The handprint concept could support addressing sustainability challenges and significantly contribute to global sustainable development targets (e.g., Agenda 2030) [5]. Handprints can play an essential role in encouraging and promoting contributions to sustainability through a focus on positive actions of organizations, individuals, and corporations [6].

Handprints are about (1) a measure for positive action, collaboration, and networking towards sustainability and a tool for measuring the positive impact of actions to promote sustainable development [6]; (2) a normative approach (what should be done and not only what has been done) [7]; (3) a measure of action by individuals who support measurable changes in behavior towards sustainable development and the environment [8]; (4) actions to improve the well-being of people or the sustainability or healing of the planet [4]; and (5) focusing on the positive ways to think about sustainability and taking appropriate action [9]. Handprint thinking is about the good we do with unlimited potential [10], structural changes to promote sustainable behavior by all people [11] and the encouragement of people to work for sustainable development [12]. Social handprints refer to (1) results of changes (as compared to business as usual) that create positive outcomes or impacts and changes that go beyond or address the organization/product value chain and create additional/unrelated positive social impacts or reduce the social footprint [13] and (2) changes to business as usual that create positive impacts [14].

Previous studies have recognized that (1) handprints can be social (e.g., reaching the living wage), environmental, and economic [3]; (2) there is a need for research on the extension of the handprint approach to more holistic sustainability handprint (taking into account social, economic, and environmental handprints of a product) [15], (3) handprints are emerging as a promising tool for promoting sustainability improvement and that more focus is needed on alternative handprint assessment approaches, including the incorporation of social science understanding of pathways and agency into assessments and methods, tools, and data sources [7]; (4) there is a need for approaches and indicator systems to address the contributions of businesses to the UN SDGs [16]; (5) the handprint approach could be extended to the development of a sustainability handprint [5]; (6) modern companies can significantly benefit from the development of the sustainability handprint concept [17]; and (7) there are many ways to assess a handprint that encompass the potential role of the handprint assessment in moving toward sustainability and the different perspectives in the world [7].

The handprint approach is closely linked to sustainability assessment, including social sustainability. For example, previous studies have recognized that (1) there are challenges in the implementation of the sustainability concept by most organizations, especially related to the determination and measurement of sustainability performance (of products/processes in particular), including, e.g., the selection/quantification of social criteria (taking into account research/consensus needs of the involved stakeholders) and overall data availability [18]; (2) there are significant theoretical and practical challenges in the development of an assessment approach to the social sustainability of products/processes due to, e.g., high complexity of the social sustainability dimension, data availability, assessment method issues, and acceptance of the approach by the public and the industry [19]; (3) social sustainability should be integrated into project life cycle management and technology through checklists and guidelines (partly due to challenges related to the application of a quantitative social impact assessment method), which may contribute to a paradigm shift in industry about obtaining and evaluating information about social impacts [20]; and (4) the social dimension plays a major role in sustainability assessment [21]. In addition, the following findings have been highlighted:

- It is important to integrate social aspects into decision processes and to combine them with other methods (even if only some aspects of social sustainability are addressed due to methodological and practical restrictions) because social impacts play a major role in sustainability assessment [22].

- Further development of sustainability assessment (of technologies) requires focusing on an appropriate and sufficient amount of (social) indicators, taking into account technology implementation conditions, whole life cycle perspective (e.g., supply chain), and lack of data and improved databases [23].
- Assessment of the social sustainability of technology and engineering projects encompasses focusing on stakeholder participation (information provision and stakeholder influence), external population (community, human, and productive capital), macrosocial performance (socio-environmental and socio-economic performance), and internal human resources (employment stability/practices, capacity development, and health/safety) [20].
- Sustainability science has not been taken into account in sustainability assessment studies [24].
- Sustainability assessments need to take into account that human needs are interlinked and intertwined with other entities (e.g., nature and resources), and integrated into the earth system and its support to well-being and the impact on human well-being (e.g., health and happiness) [25].

## 2. Materials and Methods

This study aimed at exploring, discovering, analyzing, and synthetizing the main implications of (1) handprint definitions, elements, and approaches, including application in organizations and companies; (2) handprint thinking definitions, elements, and approaches; (3) handprint approaches and applications in organizations and companies; (4) life cycle thinking, approaches, and management; (5) social life cycle assessment (S-LCA) approaches; (6) social–organizational life cycle assessment (SO-LCA) approaches; and (7) life cycle sustainability (LCSA) approaches for the creation and assessment of social sustainability handprints.

In addition, this study aimed at exploring, discovering, analyzing, and synthetizing the implications of (1) environmental, ecological, and carbon handprints for the assessment of social/society–environment relationships and interfaces in the context of social sustainability handprints; (2) challenges and limitations associated with S-LCA, SO-LCA, and LCSA approaches for the assessment of social sustainability handprints; and (3) development focus areas associated with S-LCA and LCSA approaches for the assessment of social sustainability handprints. The chosen approach is creative and innovative, and highlights novelty because there are no similar studies on social sustainability handprints.

This study applied a qualitative research approach [26] based on the idea that the research approach needs to be defined based on the purpose of the study. The following specific approaches were applied [26]: (1) collection and analysis of mostly qualitative information using textual materials; (2) inductive, deductive, and abductive reasoning; (3) organization and synthesis of information and content analysis (e.g., evaluation and critical inquiry); (4) building of a conceptual framework that evolves and changes, driven by new insights and progress of the study; (5) category and pattern construction (e.g., interrelationships, influences, and interaction); and (6) summative synthesis and statements (e.g., linkages). In addition, the chosen qualitative approach focused on open discovery, new insights/understandings, and on description, analysis, and interpretation [26].

Social and societal sustainability are often intertwined, and this was taken into account through the inclusion of an overall societal perspective in the approach. Interrelationships between social and economic sustainability including social/society–economy relationships and interfaces were out of the scope of this study even though many related social/societal aspects and contexts were addressed. The materials included scientific articles, research reports, and other publications (searched for in all major academic research databases) as well as online sources. The search was focused on handprint and life cycle thinking, approaches, and studies, with particular emphasis on social sustainability. This study is accompanied by another study (Part 2) that addresses social sustainability handprints in

the contexts of sustainability and sustainable development, including a more detailed focus on social sustainability and sustainability assessment.

This study acknowledged that there are multiple potential and useful approaches to the creation and assessment of social sustainability handprints that can be based on and linked to the comprehensive frameworks of both sustainability and sustainable development. Previous studies [3] have recognized that handprint approaches can be dynamic and qualitative, including a focus on actions [8,10,12], or static and quantitative. For example, handprint approaches include (1) actions to promote sustainability and sustainable development [8], the good we do with unlimited potential encompassing an inspiring, educating, and influencing approach as well as individual/collective creation (at home/work) [10]; (2) approaches to encourage people and individuals to work for sustainable development, including a joint effort to promote a transformation towards a sustainable society and to implement sustainability [12]; (3) a focus on positive actions and changes (e.g., innovations and initiatives) by organizations, individuals, and companies [3]; (4) positive sustainability contributions, actions, and impacts [17]; and (5) sustainability improvements [7].

In addition, handprint approaches include (1) solving societal and environmental challenges, (2) promoting positive changes, (3) supporting sustainability transformations (societies and businesses), and (4) assessment of positive contributions to sustainable development (e.g., active contributions of organizations) [16]. They can also be about innovation and collaboration among multiple actors to promote sustainable development [3], interconnected innovation, and systemic thinking approaches to sustainability (e.g., sustainability management and assessment) [4], addressing sustainability challenges and contributions to global sustainable development targets [5] and holistic approaches to sustainability [15].

Therefore, approaches to create and assess social sustainability handprints can be based on, e.g., handprint and life cycle thinking and approaches; sustainability management, assessment, and indicators (e.g., sustainability indicators/index/indices [27–30]); and sustainability science and research approaches. This means that sustainability handprints can be created through and assessed based on multiple approaches, such as (1) innovations, changes, actions/activities, initiatives, and positive impacts; (2) sustainability management and assessment (e.g., indicators, index/indices, and metrics); (3) sustainability science and research approaches; (4) handprint and life cycle thinking and approaches; (5) leadership, informed decision-making, governance, design, planning, and sustainable engineering; and (6) improvements and changes towards sustainability and sustainable development, including, e.g., social/societal sustainability and social/society-environment relationships and interfaces. For the purposes of this study, the social sustainability handprint concept and approach can be presented in a simplified manner as follows:

$$\text{Social sustainability handprint} = (\text{social sustainability}_{\text{(normal practice)}} + \text{Social sustainability handprint}_{\text{(social sustainability)}}) - \text{social sustainability}_{\text{(normal practice)}} \quad (1)$$

where social sustainability $_{\text{(normal practice)}}$ refers to the normal social sustainability practices/performance level (that can be used, e.g., as a baseline level of social sustainability) associated with, e.g., an organization, company, society/societal actor, group of people, individual(s), products/services/processes, or an activity/activities based on social sustainability assessment based on, e.g., (1) sustainability science/research approaches, (2) sustainability management and assessment using indicators/index/indices, and/or (3) handprint and life cycle thinking and approaches.

A social sustainability handprint $_{\text{(social sustainability)}}$ refers to actions, innovations, changes, impacts, and initiatives that result in the improvement of social sustainability practices/performance level associated with an organization, company, society/societal actor, a group of people, individual(s), products/services/processes, or an activity/activities based on social sustainability assessment based on, e.g., (1) sustainability science/research approaches, (2) sustainability management and assessment using indicators/index/indices, and/or (3) handprint and life cycle thinking and approaches.

### 3. Results and Discussion

*3.1. Handprints*

3.1.1. Handprint Definitions, Elements, and Approaches

There are multiple handprint definitions, elements, and approaches and they are often closely linked to or directly based on sustainability and sustainable development. The overall handprint framework is closely connected to global trends and developments related to, e.g., (1) actions that impact societal, environmental, and economic sustainability and action and changes in behavior towards sustainable development [7,8]; (2) the need to develop approaches and indicator systems to address the contributions of businesses to the UN SDGs [16]; (3) movement of companies (beyond health and safety) towards well-being within both operations and supply chain and system-level change towards sustainability, including the measurement of positive impacts on human health and the environment [4]; (4) working together for a transformation towards a sustainable society and to implement sustainability (e.g., focus on participation and institutions) [12]; (5) addressing global challenges such as climate change and biodiversity loss by making improvement opportunities visible and more reachable based on a positive approach [7]; and (6) progress towards a world in which companies develop the ability, organizational culture, and situational awareness to measure and improve the well-being of their workers [4].

In previous studies, these definitions, elements, and approaches have focused on, e.g., (1) action towards sustainability and collective/individual actions to solve environmental problems [2]; (2) positive actions by individuals to promote societal and environmental aspects of sustainability and to improve the conditions for life on the planet now and in the future [8]; (3) the creation of change towards sustainability based on active improvement and development measures [17]; (4) positive impacts of products, processes, and services of companies on the planet and people (including ripple effects of these positive actions) [4]; (5) management of corporate sustainability performance, including social responsibility, ecological balance, political participation, and economic capability [31,32]; (6) corporate social responsibility including, e.g., an objective assessment of who is doing well, supply chain ethics, and answering the question of whether you are leaving the earth a better place than you found it [33]; and (7) beneficial changes and the impacts of positive changes (relative to what would have happened without that change) [34–37].

It has been acknowledged that handprints tend to be very social because most events have multiple causes and people are all connected (footprints will eventually be reduced by the handprints of other people) and very creative, and handprints can be anywhere in the world, including in multiple small impact reductions [38]. There can be a range of different handprints and approaches to handprint assessments (depending on the conception of the concept), including various ways to address the focus (e.g., organization, individual, or product/service) and the inclusion of all improvements based on indicators, purpose, and context-specific approaches [7].

These findings suggest that social sustainability handprints can be created through multiple actions, changes, and positive impacts to promote social sustainability, including social/society–environment relationships and interfaces. In addition, they can be applied by all societal organizations and individuals encompassing all dimensions of social sustainability and all social and societal aspects of sustainable development both locally and globally. Approaches to create social sustainability handprints needs to have a strong focus on active development, innovation, creativity, and improvement as drivers for change/impacts/action towards sustainability and sustainable development. This kind of comprehensive approach allows for multiple focus and improvement areas, purposes/goals, indicators, and contexts.

Social sustainability handprint development needs to address the key elements of global development through providing (1) a way to create action, changes, and impacts towards both sustainable development and sustainability; (2) assessment approaches and indicators for the practical implementation, management, and assessment of social sustainability in all types of organizations; (3) a way to integrate all supply chain actors

into social sustainability management and assessment; and (4) approaches to address sustainability challenges and to promote more sustainable societies. Additionally, the findings of previous studies suggest that social sustainability handprints can be assessed using multiple qualitative and quantitative approaches to assess all the described ways to create the handprints at various contexts and levels (e.g., local organizations). Further implications of handprint definitions and elements for the creation of social sustainability handprints are presented in Table 1, and the implications of handprint approaches for the assessment of social sustainability handprints are presented in Table 2. All points that address environmental sustainability (e.g., on the planet level) imply that social/society–environment relationships and interfaces need to be integrated into the creation and assessment of social sustainability handprints.

**Table 1.** Implications of handprint definitions and elements for the creation of social sustainability handprints.

| Handprint Definitions and Elements | Ways to Create Social Sustainability Handprints |
|---|---|
| (1) Symbol of, commitment to (a pledge to act) and measure for positive action, collaboration and networking towards sustainability and (2) a tool for measuring the positive impact of actions to promote sustainable development [6] | (1) Organizational or personal commitment to social sustainability and (2) positive actions and impacts and collaboration and networking to promote social/societal sustainability aspects of sustainable development |
| (1) Encouragement of the creation of positive changes, actions, innovations and impacts [3] and (2) motivation by focusing on the positive ways to think about sustainability and take appropriate action [9] | Positive social sustainability changes, actions, innovations and impacts (supported by positive social sustainability vision/awareness) |
| A holistic and innovative approach to enable the measurement, evaluation and communication of the social and environmental sustainability positive impacts of organizations and products [14] | Positive social sustainability impacts of organizations/products |
| Contribution to (1) solving of societal and environmental challenges, (2) promoting positive changes within product life cycles and (3) supporting a sustainability transformation of business and society [16] | (1) Solved societal sustainability challenges, (2) positive social sustainability changes within life cycles and (3) societal and business-related social sustainability transformation |
| (1) The beneficial social and environmental impacts that we can achieve [38] and (2) actions to improve the well-being of people and/or sustainability/healing of the planet as compared to business-as-usual [4] | (1) Achieved beneficial social sustainability impacts and (2) actions to improve the well-being of people and to promote social sustainability |
| Positive action, caring, inspiring and working together towards a sustainable future including the whole planet (and all life on it) [7,39] | Positive actions, working together, caring and inspiring to promote social/societal aspects of a sustainable future |
| (1) Contributes towards a sustainable society and planet through more sustainable lifestyles, (2) can be applied in multiple ways, (3) supports the analysis of personal sustainable action and assessment of impacts on the planet and reaching out to others and (4) reflects the commitment to action, spirit of hope and enthusiasm on behalf of the global community [8] | (1) Contributions towards a sustainable society and social/societal aspects of more sustainable lifestyles, (2) personal and collective social sustainability actions and impacts and (3) social sustainability actions by the global community |
| Potential to (1) address sustainability challenges and (2) significantly contribute to global sustainable development goals (e.g., Agenda 2030) [5] | (1) Addressed social sustainability challenges and (2) promotion of social/societal aspects of global sustainable development goals |
| (1) Individual/collective creation (at home/work) and (2) a powerful positive feedback loop that can be magnified by influencing others (personal handprints can influence the actions of others over a long time period) and persistence in individual efforts [10] | (1) Individual/collective actions (at home/work) over a long time period, (2) influence on others and (3) persistence in individual efforts |

**Table 1.** *Cont.*

| Handprint Definitions and Elements | Ways to Create Social Sustainability Handprints |
|---|---|
| The positive impacts of our decisions and actions on the world (that we create beyond the boundaries of our footprint) and positive changes that we cause (that can include many positive benefits) including full ripple effects [40] | (1) The positive social sustainability impacts, decisions and actions and (2) created positive social/societal sustainability changes/benefits including full ripple effects |
| Symbol for creativity, the good we do, impact of our actions and a system changer [41] | Action and changes based on use of creativity including impacts of actions and system changes |
| (1) Positive activities that lead to structural changes (e.g., changing the social environments) and (2) an active role in changing societal structures at many levels and increasing the sphere of action (increasement of handprint) [11] | (1) Positive social sustainability activities, (2) structural social/societal sustainability changes and (3) actively changed societal structures (at many levels) and increased sphere of action |
| (1) Transformative environmental, social and economic positive changes [3], (2) innovation (changes to business-as-usual impacts) covering all sustainability impacts [4] and (3) inventions, entrepreneurship and societal/personal altruism can support the creation of handprints [10] | (1) Transformative and positive social sustainability changes, (2) social/societal innovations (changes to business-as-usual impacts) covering all social sustainability impacts and (3) social/societal sustainability inventions and entrepreneurship |
| (1) Demonstration of progress in sustainability by companies and (2) added value through encouragement of individual agency and potential increasement of the sense of empowerment [7] | (1) Progress in social sustainability by companies, (2) encouragement of individual social/societal sustainability agency and (3) increasement of the sense of empowerment |
| (1) Efforts to create a larger global handprint (agent of positive change and member of a group of actors that create positive impacts), (2) beneficial/positive environmental and social impacts that can be achieved through intentional future changes and (3) the creation, management and reporting of positive changes and impacts [17] | (1) Future-oriented positive actions/changes/impacts to promote social sustainability, (2) beneficial/positive social sustainability impacts through intentional future changes and (3) social sustainability management and reporting |

**Table 2.** Implications of handprint approaches for the assessment of social sustainability handprints.

| Handprint Approaches | Social Sustainability Handprint Assessment Approaches |
|---|---|
| A measure of action (1) by individuals to support measurable change in behavior towards sustainable development and environment and (2) of what we can do individually and together to restore the balance between the carrying capacity of the planet and consumption [8] | Measurement of individual/collective behavior change to promote social/societal sustainability aspects of sustainable development |
| Inclusion of all improvements based on indicators/purpose/context specific approaches [7] | Integration of (1) all social/societal sustainability improvements based on social sustainability indicators, (2) social/societal sustainability purpose and (3) context specific approaches into assessment, development of indicators and collection of information |
| Approaches and indicator systems to address the contributions of businesses to the UN SDGs [16] | Social sustainability indicator systems to assess the contributions of businesses to social aspects of the UN SDGs |
| (1) Social approach (most events have multiple causes), (2) connected/creative people, (3) handprints anywhere in the world and (4) assessment of the impacts of efforts to change something in the world (intentional changes to the future e.g., through projects) collectively or individually [38] | Integration of (1) multiple events and associated causes, (2) connected and creative people, (3) global scale and (4) social sustainability impacts of collective and individual efforts to change something in the world (intentional changes to the future) into assessment, development of indicators and collection of information |
| Promotion of (1) new ideas and creativity about more positive company impacts and (2) of systemic thinking within companies taking into account positive actions and impacts of company operations [17] | Integration of (1) a focus on new ideas/creativity about more positive social/societal sustainability impacts by companies and (2) systemic thinking within companies considering positive social/societal sustainability actions and impacts of company operations into assessment, development of indicators and collection of information |

**Table 2.** *Cont.*

| Handprint Approaches | Social Sustainability Handprint Assessment Approaches |
|---|---|
| (1) Encouragement of the creation and estimation of positive impacts of and (2) accounting for positive changes caused by organizations, companies and individuals (e.g actions that could lead to potential positive changes such as innovations that improve the life cycle performance of a product, investments and initiatives) [3] | Assessment of positive social/societal sustainability impacts of and accounting for positive social/societal sustainability changes caused by organizations, companies and individuals (e.g innovations and improvement of life cycle performance) |
| Life cycle assessment approaches, tools/databases to track/report environmental footprints and international standards and reporting frameworks [4] | Life cycle approaches including the use of international standards and reporting frameworks |
| (1) Creation through basic decisions, choices, events and changes in lifestyle/consumption and (2) dependence on all kinds of information (in many forms and from many sources) [38] | Integration of (1) all kinds of information (many forms/sources), (2) basic decisions, choices and events and (3) changes in lifestyle/consumption related to social/societal sustainability into assessment, development of indicators and collection of information |
| Assessment of positive contributions to sustainable development (e.g., the active contribution of organizations to sustainable development and stakeholder inclusion and education) [16] | Integration of positive social/societal sustainability contributions to sustainable development (e.g., by organizations) including stakeholder inclusion in and education into assessment, development of indicators and collection of information |
| (1) Positive approach to impact assessment that can inspire and motivate company employees, (2) creation of positive changes in the whole supply chain, (3) intentional/voluntary positive actions and changes towards sustainability (e.g., active changes to the future and measurement of the impacts of those changes) and (4) measurement/communication of the positive changes of actions and the beneficial impacts created within the life cycle of products, organizations, services, companies, processes or individuals [17] | Integration of (1) positive actions/changes towards social sustainability including active changes to the future and measurement of associated impacts, (2) beneficial/positive social sustainability impacts through intentional future changes, (3) positive social sustainability changes and impacts, (4) the positive social sustainability changes of actions and (5) the beneficial social sustainability impacts created within organizations, companies, individuals, the life cycle of products, services or processes into assessment, development of indicators and collection of information |
| (1) What is being improved including the selection of indicators based on the purpose of the handprint assessment (indicators could be linked to e.g., progress on sustainable development goals), (2) which changes will be included and from which baseline (e.g., actual handprint based on assessment of past activities or handprint potential focusing on future improvements) and (3) whose action does the handprint captures and by which pathway of influence (e.g., selection of actor such as non-governmental organization, individual, humanity, company or country based on handprint purpose and audience) [7] | Integration of (1) improved social sustainability aspects and selection of social sustainability indicators based on the purpose of the assessment (e.g., progress on social/societal aspects of sustainable development goals), (2) social sustainability changes (e.g., potential for future improvements or assessment of past activities) and (3) addressed social/societal sustainability actions (e.g., organizational, individual, company or country based on purpose/audience) into assessment, development of indicators and collection of information |
| Dynamic life cycle assessment: (1) positive actions and changes caused by an actor both within (internal handprint) and outside (external handprint) the scope of the footprint of the actor, (2) environmental, social and economic impacts and (3) every change caused by an actor (anywhere) [3] | Dynamic life cycle approaches including (1) positive social/societal sustainability actions and changes, (2) social sustainability impacts and (3) all social/societal sustainability changes globally |
| (1) Life cycle and systemic thinking and (2) continuous improvement of environmental performance (e.g., measurement of positive and reduced negative impacts) [17] | Integration of (1) life cycle/systemic thinking of social/societal sustainability and (2) continuous improvement of social sustainability performance (e.g., measurement of positive social/societal sustainability impacts) into assessment, development of indicators and collection of information |

Previous studies have also focused on specific handprints such as environmental, ecological, and carbon handprints. Ecological handprints are about (1) ideas that support both people and the planet and (2) innovative and robust solutions that address poverty and climate change at the same time [42]. The metrics of the environmental handprint can

encompass the metrics of accomplishment (e.g., a park), small steps that promote major outcomes (e.g., technology innovations and experiments), pilot projects, demonstrations, and even failed programs if they provide significant learning [10]. Carbon handprints can be used by organizations to demonstrate and compare positive climate impacts [15]. These findings suggest that the creation and assessment of social sustainability handprints need to take into account social/society–environment relationships and interfaces. The implications of environmental and ecological handprint definitions, elements, and approaches for the assessment of social/society–environment relationships and interfaces in the context of social sustainability handprints are presented in Table 3, and the similar aspects and implications of carbon handprints are presented in Table 4.

**Table 3.** Implications of environmental and ecological handprint definitions, elements, and approaches for the assessment of social/society–environment relationships and interfaces in the context social sustainability handprints.

| Definitions, Elements, and Approaches | Approaches to the Assessment of Social/ Society–Environment Relationships and Interfaces |
|---|---|
| The good we do for the environment (environmental handprint) with unlimited potential and no limit to the good one can do [10] | Integration of all the good we do including unlimited potential into assessment, development of indicators and collection of information |
| Ecological handprints that (1) take place at the interface of social justice and environmental restoration, (2) are market driven and locally controlled solutions to economic poverty and environmental ruin, and (3) lift humanity and communities out of poverty and lower ecological footprint at the same time [42] | Integration of (1) social justice and environmental restoration, (2) locally controlled solutions, and (3) community level aspects into assessment, development of indicators, and collection of information |
| Environmental handprint engages the power of creativity, idealism, and profit, and the creation of an environmental handprint is more about creating an opportunity than being given an opportunity [10]. | Integration of creativity and creation of opportunities related to social/society–environment relationships and interfaces into assessment, development of indicators, and collection of information |
| Ecological handprints highlight how communities and entrepreneurs (in the developing world) are achieving the global vision (based on the UN SDGs) of ending poverty, protecting the planet, and ensuring that all people enjoy prosperity and peace through ecological and profitable solutions (focus on turning global goals into local reality) [43]. | Integration of local achievement of relevant aspects of the UN SDGs by communities/entrepreneurs into assessment, development of indicators, and collection of information |
| Ecological handprints that (1) expand upon the ecological footprint by linking together the interrelated goals of ensuring sustenance for those in need and sustaining the biological integrity of the planet (recognizing the importance of the interrelationship between these goals), (2) are designed to encompass or measure impacts on human development or humanitarian issues (e.g., social justice, human rights, and poverty), and (3) are a problem-solving approach based on a wide range of innovative efforts that improve human well-being and have a low footprint [44] | Integration of (1) approaches to ensure the sustenance of people (in need) and to sustain biological integrity, (2) the interrelationship between human and planetary goals, (3) impacts on human development/humanitarian issues (e.g., social justice, human rights, and poverty), and (4) problem-solving approaches based on multiple innovative efforts to improve human well-being and sustainability into assessment, development of indicators, and collection of information |

**Table 4.** Implications of carbon handprint definitions, elements, and approaches for addressing social/society–environment relationships and interfaces in the context of social sustainability handprints.

| Definitions, Elements, and Approaches | Approaches to the Assessment of Social/ Society–Environment Relationships and Interfaces |
|---|---|
| (1) Support of strategic decision-making and long-term climate goals related to the production and use of solutions, (2) extending the environmental responsibility of companies beyond their gates (e.g., enhancement of value chain cooperation), and (3) a tool for companies to manage their climate impacts, including focus on product assessment, taking into account use by customers and placing the product in the surrounding environment [5] | Integration of (1) strategic decision-making, (2) long-term climate goals, (3) value chain cooperation, and (4) management/assessment of climate impacts into assessment, development of indicators, and collection of information |

**Table 4.** *Cont.*

| Definitions, Elements, and Approaches | Approaches to the Assessment of Social/ Society–Environment Relationships and Interfaces |
|---|---|
| Creation by organizations through multiple pathways, including (1) extending the lifetime and improvement of the performance of products, (2) carbon storage and capture, and (3) recycling, reusing, and remanufacturing [5,15,45] | Integration of (1) product lifetime/performance, (2) carbon storage/capture, and (3) recycling, reusing and remanufacturing into assessment, development of indicators, and collection of information |
| Benefits related to climate change, e.g., through comparing the beneficial actions against business as usual [17] | Integration of beneficial climate change actions into assessment, development of indicators, and collection of information |
| Use by organizations to demonstrate the positive climate impacts provided by their product to potential customers and to compare their products to baseline products [15] | Integration of climate impacts of organizations into assessment, development of indicators, and collection of information |
| (1) Climate change mitigation potential associated with customer activities due to replacement of a baseline solution with a handprint solution by the customer [45], (2) the reduction of the carbon footprint of customer/customers [5,15], and (3) the GHG reductions of the customer that the product enabled [15] | Integration of climate change mitigation potential of activities by actors (e.g., stakeholders of organizations) into assessment, development of indicators, and collection of information |
| Use by organizations to support decision-making, lifelong product design [5,15], and communication and marketing [15] | Integration of decision-making and design by organizations into assessment, development of indicators, and collection of information |
| (1) Communication of climate benefits of technologies, products, and services; (2) identification of opportunities to improve the climate performance of products and potential development needs; (3) informing/advising decision-makers/stakeholders; and (4) support of product development and strategic/political decision-making [45] | Integration of climate benefits/improvements and decision-making into assessment, development of indicators, and collection of information |

### 3.1.2. Handprint Thinking

Handprint thinking is about the broad theoretical, conceptual, and framework building elements behind the handprint approach that highlight the significant potential and possibilities related to this comprehensive approach, including social sustainability handprints. Previous studies have recognized that (1) handprint thinking can be applied over time (broadens the range of responses to both social and environmental challenges), shared handprint thinking can translate analysis into action, and collective handprint is much more than the sum of all individual efforts (indicates very high potential) [10]; (2) changes to the future and guiding actions that make people and organizations a net benefit to the world (bringing more to the world than taking from it), encompassing the same comprehensive set of sustainability-related impacts (e.g., social and environmental impacts) [38]; (3) encouragement of people to work for sustainable development [12]; (4) a normative approach, e.g., addressing the issue of what should be done (not only what has been done) [7]; and (5) measurement of well-being [4].

These findings suggest that both the creation and assessment of social sustainability handprints need to be future oriented based on holistic/comprehensive approaches to social sustainability and well-being (including associated indicators). Focus is needed on broader collective actions (e.g., organizational) and individual efforts, various contributions of people towards sustainable development, and normative approaches. Further implications of handprint thinking definitions and elements for the creation of social sustainability handprints are presented in Table 5, and the implications of handprint thinking approaches for the assessment of social sustainability handprints are presented in Table 6. All points that address environmental sustainability imply that social/society–environment relationships and interfaces need to be integrated into the creation and assessment of social sustainability handprints.

**Table 5.** Implications of handprint thinking definitions and elements for the creation of social sustainability handprints.

| Handprint Thinking Definitions and Elements | Ways to Create Social Sustainability Handprints |
|---|---|
| (1) The good we do; (2) unlimited potential; (3) inspiring, educating, and influencing approach; (4) recovery and restoration; (5) keeping count of accomplishments; (6) entrepreneurism; (7) advocation for protection; and (8) appreciation and celebration [10] | (1) The good we do to, (2) education, (3) inspired/influenced people, and (3) entrepreneurship to promote and manage social sustainability with unlimited potential |
| (1) Changing structures to promote sustainable behavior by all people [11]; (2) encouragement of people to work for sustainable development, sustainable behavior; and to use social scope in design [12]; and (3) active creation of social and political changes and focus on enhanced social/political commitment to promote sustainability [46] | (1) Structural changes to promote and work for social/societal aspects of sustainable behavior and sustainable development by all people, (2) social sustainability scope in design, and (3) social/societal changes |
| (1) Support of informed decision-making, (2) inspiring others (e.g., handprinting community), and (3) reducing/counterbalancing footprints [40] | Informed decision-making, inspiration of others, and community-level efforts to promote social/societal sustainability |
| (1) Changes to the future and guiding actions that make people and organizations a net benefit to the world (bringing more to the world than taking from it), (2) the same comprehensive set of sustainability-related impacts (e.g., social impacts such as human rights, poverty, community impacts, and working conditions and environmental impacts such as ecosystem quality, climate change, and resource depletion), and (3) full life cycle/supply chain consequences of actions [38] | (1) Changes to the future and actions that ensure the promotion and management of social sustainability by organizations/people (to bring more social/societal sustainability to the world than taking from it), (2) comprehensive set of social sustainability impacts, and (3) full life cycle/supply chain social/societal sustainability consequences of actions |
| The impact of a single handprint (e.g., planting a tree) can be self-renewing and it is theoretically unlimited [41] | Unlimited and self-renewing impacts of single actions to promote social/societal sustainability |
| Provision of an important and useful perspective even without actual handprint assessments [7] | Developing new perspectives on social sustainability that lead to action, changes, innovations, and positive impacts |

**Table 6.** Implications of handprint thinking approaches for the assessment of social sustainability handprints.

| Handprint Thinking Approaches | Social Sustainability Handprint Assessment Approaches |
|---|---|
| Individual and collective approaches (e.g., environmental educators can create handprints by influencing future generations) [9] | Integration of collective/individual actions (e.g., education of future generations) to promote social/societal sustainability into assessment, development of indicators, and collection of information |
| (1) Measurement of well-being and (2) creation of handprints to benefit people via companies using better tools to assess current needs and to measure success (well-being is about subjective rating of the quality of life based on personal experience of purpose, social connection, learning, mastery and self-efficacy, and personal satisfaction with general welfare, health, safety, and security) [4] | Integration of well-being and associated human needs based on a comprehensive set of social/societal sustainability indicators (e.g., subjective and collective assessment of quality of life) into assessment, development of indicators, and collection of information |
| Assessment of the impact of changes made by value chain actors (companies/suppliers) to improve their social impacts (to grow social handprint and reduce social footprint) [13] | Integration of social sustainability impacts of changes made by value chain actors (e.g., companies/suppliers) into assessment, development of indicators, and collection of information |
| (1) Planning (plan) and implementation (change) of a change (intervention) and (2) measurement of the associated results (social handprint creation and social footprint reduction) [14] | Integration of planning, implementation, and changes/interventions to promote social sustainability into assessment, development of indicators, and collection of information |

| Handprint Thinking Approaches | Social Sustainability Handprint Assessment Approaches |
|---|---|
| (1) Changes achieved through voluntary/intentional action [37]; (2) creation of positive benefits [35], including in relation to business as usual [37]; (3) creation of positive impacts [36]; and (4) avoiding/preventing footprints [35], reducing total footprints relative to business as usual [37], and being a cause of reductions in the footprint of some other actor relative to business as usual [36] | Integration of (1) changes to promote social sustainability through voluntary/intentional action, (2) creation of positive social sustainability benefits/impacts, and (3) avoiding/reducing/preventing negative social sustainability impacts (collectively/individually) into assessment, development of indicators, and collection of information |
| (1) Normative approach (e.g., what should be done and not only what has been done), (2) encouragement of and addressing positive impacts against a certain baseline, and (3) going beyond current footprint accounting practices, e.g., by measuring different things such as positive impacts or impacts of others or by focusing on how/what action will be taken in practice (and by who, when, and where) [7] | Integration of (1) normative aspects (what should be done) to promote social sustainability, (2) promotion of positive social sustainability impacts (e.g., against some baseline), and (3) going beyond current social sustainability practices (e.g., different things, positive impacts, practical actions, and impacts of others) into assessment, development of indicators, and collection of information |
| (1) Measurement and reduction of own footprints; (2) helping, empowering, and incentivizing the footprint reductions of others (e.g., supply chain innovations); and (3) taking generative actions addressing the same types of impact categories for which footprints cause harm (e.g., tree planting and promotion of lifestyles of employees/families/communities) [4] | Integration of (1) actions to reduce negative social sustainability impacts (e.g., organizational and supply chain) and (2) supply chain social sustainability innovations into assessment, development of indicators, and collection of information |
| (1) inspiring, educating, and influencing approach and (2) keeping count of accomplishments [10] | Integration of inspiration, education, influence on others, and multiple ways to count accomplishments into assessment, development of indicators, and collection of information |

### 3.1.3. Handprints in Organizations and Companies

All types of organizations and companies can create and use handprints for multiple purposes and in multiple contexts and applications. Organizations can create internal (e.g., promote meaning, purpose, and training at the workplace/among employees) and external (e.g., community engagement and good reputation and transparency) handprints [40] and handprints can support sustainability management and assessment [17]. Organizations and companies can take actions that could lead to potential positive changes (e.g., internal or external innovation that enables or requires innovation by a supply chain actor) [3]. It has been noted that organizations are both the solution and the problem and that consequently they need to be inspired, persuaded, lobbied, educated, and even compelled to act [33]. Handprints can motivate and inspire positive changes and impacts locally, nationally, and globally, including their contribution to global sustainable development goals, and handprints can be created through new innovations, solutions, products, and services based on active development and improvement measures to promote positive impacts [17].

Handprints can support (1) sustainability management (whole value chain), assessment, and performance measurement; (2) informed decision-making and strategic management; (3) product or process design/redesign; (4) change management; (5) reporting; and (6) marketing, product declaration, and certification/labelling [17]. Handprints can help companies to become more sustainable, including positive contributions (e.g., to society, employees, and customers) [17] and to consider the broad set of actors/activities within a complex global operational environment and international market encompassing a network of multiple supply/value chain actors [17]. They can also be used in the management of global corporate sustainability performance and associated sustainability dimensions, including specific fields of action (e.g., social responsibility, political participation, and ecological balance) as a basis for specific measures [31,32].

Previous studies have recognized that handprints can (1) re-energize the sustainability actions and intentions of companies/individuals and expand organizational missions (and the meaning of our lives) considering the principle of co-creative relationships [4];

(2) inspire, organize, and reward local positive initiatives and support of long-term sustainability [33]; (3) broaden the scope of corporate sustainability (taking into account the positive actions and impacts of company operations during the full life cycle of their products), including continuous improvement of performance (e.g., voluntary innovations) and actions (e.g., management, planning, reporting, and product development) [17]; and (4) be integrated into sustainability management and reporting practices of companies, including measurement approaches, databases, and high-quality materials [17].

These findings suggest that the creation and assessment of social sustainability handprints can be integrated into organizational practices through (1) sustainability management and assessment (e.g., performance level), (2) informed decision-making and strategic management, (3) change management, (4) design and redesign of products and processes, and (5) reporting and declarations/certification. In addition, organizations need to be engaged in continuous learning/training activities related to social sustainability and to promote the creation of both internal and external (e.g., local communities and people) social sustainability handprints. The creation of social sustainability handprints through positive changes, actions, and impacts and new innovations, services, solutions, and products can promote social sustainability at many levels (e.g., help to implement global sustainable development goals).

It is important to consider and include all of society and all actors in both the creation and assessment of social sustainability handprints. A similar approach is also needed in the management and assessment of social sustainability in supply chains and among various suppliers and actors. Each individual actor can create and assess handprints and/or it can be done collectively. Comprehensive approaches require focus on systems (e.g., ecosystems of actors), networks, relationships, collaboration, collective actions, and co-creation of information and knowledge. Appropriate assessment requires qualitative and quantitative approaches, and high-quality information/data from multiple sources. Novel communication and reporting practices can be used to support both creation and assessment approaches. Further implications of handprint approaches and applications in organizations and companies for the creation of social sustainability handprints are presented in Table 7.

**Table 7.** Implications of handprint approaches and applications in organizations and companies for the creation of social sustainability handprints.

| Handprint Approaches and Applications | Ways to Create Social Sustainability Handprints |
| --- | --- |
| Creation of internal and external handprints through (1) handprint ideas (e.g., innovations, products, or services, including employee ideas), (2) encouragement/informing of other people/organizations, (3) growing the handprinting community (e.g., citizens, customers, employees, and suppliers), (4) provision of know-how, resources, and technical assistance to other organizations (e.g., community and supply chain), (5) large-scale provision of a good/service that saves money and reduces footprint, and (6) taking special and purely positive actions (e.g., tree planting) [40] | Internal/external social sustainability handprints through (1) social sustainability ideas and innovations, (2) encouragement of other people/organizations to promote social sustainability and informing them about it, (3) community-level social/societal sustainability actions (e.g., society, customers, employees, and suppliers), (4) provision of know-how and resources to other organizations (e.g., community and supply chain) to promote social sustainability, (5) social sustainability of goods/services, and (6) special/positive actions to promote social sustainability |
| Actions that could lead to potential positive changes, such as internal or external (enabling or requiring innovation by a supply chain actor) innovations, investments, and initiatives [3] | Actions to promote positive social sustainability changes including internal/external (whole supply chain) innovations, investments, and initiatives |
| (1) Positive changes/benefits created by organizations/people, (2) a wide set of pathways for positive influence, and (3) global influence by companies/production through many pathways (e.g., create sustainability benefits, sharing innovations, and inspiring action anywhere in the world) [34] | (1) Positive social sustainability changes/benefits created by organizations/people, including a wide set of positive influence pathways, (2) global social sustainability influence created by companies/production, and (3) global benefits, shared innovations, and inspired actions |

| Handprint Approaches and Applications | Ways to Create Social Sustainability Handprints |
| --- | --- |
| Potential changes influenced by (1) companies in the impacts and consumption of other organizations/individuals (e.g., in the supply chain or in communities) and (2) individuals in the impacts and consumption of other individuals/organizations (e.g., workplace or community organizations) [38] | Influenced social sustainability changes by companies on the impacts of other organizations/individuals (e.g., supply chain/communities) and by individuals on the impacts of other individuals/organizations (e.g., community organizations/workplace) |
| Creation of product related handprints through (1) improvement of the life cycle performance of an existing product through innovation, (2) introduction of a new product with better performance, and (3) increase in demand for an existing product with better performance than other products [37] | Improvement of the life cycle performance (innovation) and introduction of new products with better social sustainability performance |
| Creation through product-related innovations that can bring benefits for many impact categories (e.g., human health and ecosystem impacts) [37] | Product-related social sustainability innovations (e.g., benefits for many social sustainability impacts) |
| Positive changes due to company influence on impacts of individuals/other companies [17] | Positive social sustainability changes due to company influence on impacts of other companies/individuals |
| Voluntary reductions (relative to business as usual) in their own footprint or in the footprints of others by organizations/individuals [3] | Voluntary reductions (relative to business as usual) of negative social/societal sustainability impacts (caused by own operations or by other organizations/individuals) |

*3.2. Life Cycle Thinking and Approaches*

3.2.1. Life Cycle Thinking, Approaches, and Management

Life cycle thinking is about understanding the social, environmental, and economic impacts associated with, and the integration of, sustainability into decision-making within both private and public sectors (e.g., products, policies, services, and procurement) [47]. It is essential for sustainable development [48] and it provides a way to incorporate sustainable development into decision-making processes through its application to the pillars of sustainability [49]. There are multiple life cycle approaches that can be applied in all sectors [50], and life cycle thinking is operationalized through life cycle management (connects multiple operational concepts and tools) [50,51]. For example, industries can gain organizational benefits from integrating life cycle thinking, approaches, and management as well as sustainability management into overall management, including the development of more sustainable products and processes [52].

These findings suggest that life cycle thinking can be used to support the creation and assessment of social sustainability handprints through in-depth organizational understanding of various social sustainability impacts and the integration of social sustainability and social/societal aspects of sustainable development into decision-making, management, and assessment. Life cycle thinking and approaches can be implemented based on life cycle management, which can include life cycle approaches to social sustainability management (e.g., creation of handprints) and assessment (e.g., of handprints). Further implications of life cycle thinking and approaches for the creation of social sustainability handprints are presented in Table 8.

**Table 8.** Implications of life cycle approaches for the assessment of social sustainability handprints.

| Life Cycle Thinking | Ways to Create Social Sustainability Handprints |
| --- | --- |
| Going beyond the traditional focus on production site and manufacturing processes to include environmental, social, and economic impacts of a product, covering its whole life cycle [53] | Social sustainability impacts of a product, covering its whole life cycle (beyond the production site and manufacturing processes) |

**Table 8.** *Cont.*

| Life Cycle Thinking | Ways to Create Social Sustainability Handprints |
|---|---|
| (1) Essential for sustainable development [48] and (2) a way to incorporate sustainable development into decision-making processes (application to the pillars of sustainability) [49] | Actions, changes, and informed decision-making to promote social/societal aspects of sustainable development |
| Potential to facilitate links between social, environmental, and economic dimensions within an organization, covering its whole value chain [53] | Sustainable social/society–environment relationships and interfaces within an organization (whole value chain) |
| Integration of sustainability into and understanding the social, environmental, and economic impacts associated with decision-making (private and public sectors), including products, policies, services, and procurement [47] | Positive social/societal sustainability impacts through public/private sector decision-making (e.g., policies, products/services, and procurement) |
| Reduction of product-related resource use and emissions and improvement of its socio-economic performance, covering its whole life cycle [53] | Improvement of product-related social sustainability performance (whole life cycle) |
| Contribution to sustainability science that promotes integrated, comprehensive, and participatory approaches [54] | Contributions to sustainability science (e.g., comprehensive, integrated, and participatory approaches) |
| Progress towards sustainability, including the mainstreaming of life cycle thinking to support strategic policy in addition to product development [55] | (1) progress towards social sustainability, (2) implementation of strategic policies, and (3) product development |
| Life Cycle Approaches | Ways to Create Social Sustainability Handprints |
| (1) Support of governments in balancing and ensuring environmental, social, and economic benefits to society and (2) promotion of more sustainable consumption (e.g., through better information and public/multi-stakeholder involvement) [52] | (1) Social benefits to society (in a balanced way), (2) better social/societal sustainability information, and (3) public/multi-stakeholder involvement in actions/changes |
| Multiple approaches that can be applied in all sectors [50] | Application of similar actions and changes in all relevant sectors |
| (1) development to promote the achievement of sustainability goals and (2) proactive enhancement of positive impacts (beyond comparison of alternatives and avoidance of negative impacts) [54] | (1) The achievement of social/societal sustainability goals and (2) proactive enhancement of positive social/societal sustainability impacts (beyond comparison of alternatives and avoidance of negative impacts) |
| Making long-term-oriented choices (implies that each actor in the whole product life cycle has a responsibility/role) about, e.g., product/service design, more sustainable consumption choices by individuals, or government policies, taking into account all relevant impacts on the society, environment, and economy [52] | (1) Long-term-oriented choices, including the roles of each actor in the whole product life cycle, and (2) all relevant positive societal/social sustainability impacts |
| Comprehensive consideration of the impacts of all life cycle stages in decision-making by companies, governments, and citizens about management strategies, policies, and production/consumption [52] | (1) Positive social sustainability impacts of all life cycle stages and (2) informed decision-making by companies, governments, and citizens (e.g., produced management strategies, policies, and ways of production) |

Life cycle management can help all types of companies and organizations to operationalize life cycle thinking in practice and to promote continuous improvement of sustainability, including the whole value chain [48,50,51,56]. In addition, it provides a framework for the management and analysis of sustainability performance (e.g., goods and services) that can be used to (1) achieve sustainable development based on long-term value creation and (2) promote sustainability performance of both companies (e.g., design for sustainability and sustainable production) and value chains [56]. Previous studies have recognized that life cycle management (1) is about a dynamic/voluntary process that incorporates social, environmental, and economic aspects of products and addresses the minimization of socio-economic and environmental burdens associated with products, covering their full life cycles and value chains [48], and (2) requires more support and inputs

from social scientists and the inclusion of the values of both stakeholders and researchers (e.g., in the context of evaluating current life cycle-based sustainability initiatives) [57].

These findings suggest that life cycle management can contribute to the creation of social sustainability handprints through (1) continuous management and improvement of social sustainability performance, (2) implementation of sustainable development goals, (3) design for social sustainability, (4) actions/changes/innovations within whole life cycles, and (5) more sustainable social/society–environment relationships and interfaces. The creation of social sustainability handprints can also be supported through the integration of social science/scientists and values of stakeholders and researchers into life cycle management, including emphasis on the critical evaluation of prevailing/current life cycle approaches to social sustainability management.

In addition, life cycle management can support the assessment of social sustainability handprints through the assessment of (1) social sustainability performance; (2) social/societal aspects of sustainable development; (3) design, actions, changes, and innovations to promote social sustainability, covering whole life cycles and all value/supply chain actors; and (4) the sustainability of social/society–environment relationships and interfaces. Social sustainability handprint assessment can also be supported through the integration of assessment approaches and indicators developed within social sciences/sustainability science and by social scientists (considering the values of stakeholders and researchers) into life cycle management, including a focus on the development and critical evaluation of prevailing/current life cycle-based assessment approaches to social sustainability management.

### 3.2.2. Social Life Cycle Assessment and Social Organizational Life Cycle Assessment

Social life cycle assessment (S-LCA) aims at assessing the social impacts of products and services, covering their whole life cycles (including, e.g., supply chains), and it can contribute to an improvement in the social performance of organizations and the well-being of stakeholders [13]. Societal LCA of products, including assessment of societal impacts based on many hundreds of specific indicators, can support movement towards and monitoring of sustainability and link policy/policymakers into sustainable development [58]. In addition, S-LCA provides a means to assess the social sustainability of products and processes considering, e.g., society, employees, customers, suppliers, future generations, and the international community as stakeholders [19]. The S-LCA framework is linked to and can support the achievement of the UN SDGs based on a stakeholder approach (consideration of impacts on stakeholder categories) [13].

S-LCA can provide information on social aspects to support decision-making based on the idea that social sustainability is about the identification and management of positive/negative impacts on people/stakeholders [13]. Additionally, it can help to identify social indicators and subcategories based on social aspects of sustainability and social impact assessment literature to support decision-making, taking into account that appropriate indicators need to be adapted to specific contexts [22]. The S-LCA approach is called social organizational life cycle assessment (SO-LCA) when it is applied to organizations [14], and the SO-LCA approach can be used, e.g., to (1) compile and evaluate social and socioeconomic aspects and positive and negative impacts of organizational activities (as a whole or in part) from the life cycle perspective, (2) measure social indicators or impacts on the organizational level, (3) assess social performance of organizations (beyond the product perspective), and (4) improve social performance of organizations (focus on company-level decisions, e.g., about supplier selection/development) [13,59]. In addition, S-LCA approaches can have multiple benefits for all types of organizations, such as the following:

- Application by companies to identify potential social handprints (changes to business as usual that create positive impacts) and to get an understanding of their supply chain social footprint (negative impacts) [14];
- Assessment of social/sociological aspects of products based on qualitative, semi-quantitative, or quantitative approaches and the use of site-specific/generic data

(including actual, potential, positive, and negative impacts), covering their whole life cycles [60,61];

- Focus on society, local communities, workers, and value chain actors, including the application of specific indicators [62] and assessment of potential and verified social impacts within product life cycles to inform on the improvement of social conditions of production [63];
- Support of informed decision-making, including all social impacts of product life cycles, and focus on action to implement identified improvements [64];
- Assessment of (1) social impacts of goods and services, covering whole life cycles based on multiple indicators suitable for specific contexts, and (2) causes of improvement and reduction in well-being (promotion of social welfare in modern societies) [65].

There are multiple different S-LCA approaches that embrace the complexity associated with the measurement of social impacts and consider different intended uses and the quality of site-specific data [66]. Approaches focus on social performance, including principles, practices, and outcomes of the relationships of businesses with organizations, people, societies, communities, institutions, and the earth, covering both intended actions of businesses towards these stakeholders and unintended externalities associated with business activities [13,67]. In addition, S-LCA is about the development of life cycle thinking towards a more useful tool in the achievement of the goal of sustainable development, including the use of qualitative, quantitative, or semi-qualitative social indicators [68]. It has been recognized that (1) society; (2) companies; (3) local, national, and/or international community; (4) workers; (5) children; (6) consumers; (7) value chain actors; and (8) future generations can be used as stakeholder categories/subcategories [13,61,68].

S-LCA and SO-LCA approaches can contribute to the creation of social sustainability handprints in multiple ways, including (1) improvement of social performance of all types of organizations through management, including multiple actions, changes, positive impacts, activities, and initiatives; (2) implementation of the social and societal sustainability aspects of the UN SDGs and sustainable development; (3) informed decisions on social sustainability; (4) positive changes, actions, innovations, and impacts related to societies, employees, suppliers, the international community, and future generations; (5) qualitative and quantitative changes, impacts, and actions; (6) the use of high-quality information (e.g., site-specific) as a basis for positive actions and changes; (7) positive changes and actions focusing on the society, local communities, employees, and value/supply chain actors; (8) improvement of social conditions of manufacturing covering, e.g., potential and verified social impacts; (9) product and/or service innovations; and (10) changes and actions to improve well-being. In addition, they can support the assessment of social sustainability handprints through the following ways:

- The assessment of social performance of all types of organizations based on qualitative and quantitative approaches;
- Development and application of multiple qualitative, semi-qualitative, and quantitative social sustainability indicators;
- Use of site-specific and generic information/data, including actual, potential, positive, negative, and verified impacts;
- Integration of social sustainability into decision-making and assessment of progress towards or away from the UN SDGs;
- Provision of (1) high-quality information, (2) ways to advance social and societal well-being, and (3) a broad set of context-specific social sustainability indicators (e.g., society, local communities, employees, and supply/value chain actors);
- Integration of (1) society; (2) companies; (3) local, national, and/or international community; (4) workers; (5) children; (6) consumers; (7) value chain actors; and (8) future generations into assessment, indicator development, and information collection.

Further implications of S-LCA approaches for the assessment of social sustainability handprints are presented in Table 9, and the implications of SO-LCA approaches for the assessment of social sustainability handprints are presented in Table 10.

**Table 9.** Implications of S-LCA approaches for the assessment of social sustainability handprints.

| S-LCA Approaches | Social Sustainability Handprint Assessment Approaches |
|---|---|
| (1) Social dimension in sustainability assessment, (2) no agreed social impact assessment method, (3) social impact referring to actual experiences of an individual or community, and (4) social performance associated with the presentation of indicator results (social aspects) using certain criteria [21] | Integration of (1) social and societal sustainability dimensions, (2) multiple ways to assess social sustainability impacts, (3) social sustainability impacts as actual individual/community experiences, and (4) social sustainability performance based on social sustainability criteria and indicators into assessment, development of indicators, and collection of information |
| Assessment of social and socio-economic impacts of organizations, products, countries, and consumer supply chains and life cycles from the full life cycle perspective [14] | Integration of social sustainability impacts of organizations, products, countries, and supply chains/life cycles from the full life cycle perspective into assessment, development of indicators, and collection of information |
| (1) Usability and use in decision-making to create direct effects, (2) focus on the consequences of decisions (implemented and non-implemented) related to products to create direct effects on the stakeholders associated with the product life cycle, and (3) inclusion of social impacts on individuals taking place both in product life cycles and all aspects of their life [69] | Integration of (1) usability/use in decision-making to ensure direct effects, (2) the consequences of decisions (implemented and non-implemented) related to products, (3) direct effects on stakeholders associated with product life cycles, and (4) social sustainability impacts on individuals, covering all aspects of their lives and full product life cycles, into assessment, development of indicators, and collection of information |
| (1) Society, local community, and employees as the main stakeholder groups; (2) provision of information about the potential social impacts on people; (3) support of informed decision-making by companies about social impacts covering whole product life cycles; (4) the Universal Declaration of Human Rights as the normative basis jointly with local or national socio-economic development goals; and (5) development of indicators based on the conventions and recommendations of the International Labour Organization [70] | Integration of (1) society, local community, and employees; (2) the Universal Declaration of Human Rights; (3) local and/or national social and societal aspects of sustainable development goals; (4) the potential impacts on people (life cycle activities that affect people) due to activities in the life cycles of products; (5) informed decision-making by organizations/companies about impacts covering whole product life cycles; and (6) social sustainability indicators based on the conventions and recommendations of the International Labor Organization into assessment, development of indicators, and collection of information |
| (1) Social well-being and social justice through the creation of a positive outcome that is meaningful for societies and people, (2) well-being as the main area of protection in the assessment of social impacts of products, and (3) equity and equality addressed in terms of social justice to ensure a fair/ethical society [71] | Integration of (1) social well-being and social justice based on positive outcomes that are meaningful for societies/people, (2) well-being related to the impacts of products, and (3) equity, equality, and aspects related to fair/ethical societies into assessment, development of indicators, and collection of information |
| (1) Local community and workers as stakeholders associated with the most risk of negative social impacts [72] and (2) assessment of use-phase impacts based on literature review, focusing on social indicators (large diversity and variety) that were allocated to stakeholder groups and selected based on relevance to the study purpose [73] | Integration of (1) local communities and workers as stakeholders, (2) use-phase social sustainability impacts based on literature review, (3) social sustainability indicators considering multiple stakeholders, and (4) selection of social sustainability indicators based on relevance to assessment purpose into assessment, development of indicators, and collection of information |
| (1) The impacts of a product on stakeholder well-being, covering the whole product life cycle (implies that well-being needs to be defined), and (2) relevant social impacts identified based on participatory processes (involving affected stakeholders and their definition of what influences their well-being and in what ways), social theories about human well-being, and international conventions and standards [74] | Integration of (1) product impacts on stakeholder well-being (whole life cycle and based on comprehensive definition of well-being) and (2) the identification of relevant social and societal sustainability impacts based on participatory processes involving affected stakeholders and their definition of what influences their well-being and in what way), (3) research on human well-being, and (4) international conventions/standards into assessment, development of indicators, and collection of information |

**Table 9.** *Cont.*

| S-LCA Approaches | Social Sustainability Handprint Assessment Approaches |
|---|---|
| (1) Addressing fundamental labor rights and (2) obligatory impact categories (considered fundamental by the International Labour Organization), such as discrimination, forced/child labor, collective bargaining, and restrictions of freedom of association/right to organize [75,76] | Integration of fundamental labor rights and the main impact categories based on international (e.g., International Labour Organization) priorities, including, e.g., discrimination, forced/child labor, collective bargaining, and restrictions of freedom of association/right to organize into assessment, development of indicators, and collection of information |
| (1) Local community/society as stakeholder groups, (2) application of qualitative and quantitative surveys and direct interviews, and (3) monitoring of activities [77] | Integration of (1) local community/society as stakeholders, (2) qualitative and quantitative surveys and direct interviews, and (3) monitoring of activities into assessment, development of indicators, and collection of information |
| (1) Support of decision-making about sustainability, taking into account the whole value chain; (2) the applied indicators being linked to the UN SDGs (to allow policy/industry decision-makers to link performance to each social aspect); and (3) application of a participatory approach to determine the importance of each subcategory (local community, workers, consumers, and value chain actors) except society [78] | Integration of (1) decision-making about social sustainability (whole value chain), (2) indicators linked to social/societal aspects of the UN SDGs (to allow policy/industry decision-makers to link performance to each social sustainability aspect), and (3) participatory approaches to determine the importance of each subcategory (e.g., local community, workers, and value chain actors) into assessment, development of indicators, and collection of information |
| Numerous approaches [79] including (1) social reporting, database development, and data source sharing/provision within the supply chain and business partners; (2) identification of the main social improvement opportunities, impacts, and risks (e.g., within the supply chain); and (3) social perspective on understanding the opportunities and risks associated with the initial production process development phase [80] | Integration of (1) multiple ways to address, define, measure, and understand social sustainability; (2) social sustainability reporting; (3) development of information/data sources and bases; (4) information sharing/provision within the supply chain and business partners; and (5) social perspectives on opportunities and risks associated with the initial production process development phase into assessment, development of indicators, and collection of information |
| The social categories and associated social themes, including (1) human rights, (2) labor rights and decent work, (3) community infrastructure, (4) governance, and (5) health and safety [14,62,81] | Integration of social categories and associated social themes such as (1) human rights, (2) labor rights and decent work, (3) community infrastructure, (4) governance, and (5) health and safety into assessment, development of indicators, and collection of information |
| Application (1) to initial social sustainability screening [82] and (2) of participative approaches (fundamentally different from environmental life cycle assessment) [83] | Application in initial social sustainability screenings and the integration of participative approaches into assessment, development of indicators, and collection of information |

**Table 10.** Implications of SO-LCA approaches for the assessment of social sustainability handprints.

| SO-LCA Approaches | Social Sustainability Handprint Assessment Approaches |
|---|---|
| (1) An organizational approach to promote the improvement of both social conditions and overall socio-economic performance of an organization and its value chain (also for its stakeholders); (2) use of the benefits of life cycle-based social assessments of organizations/products to promote the improvement of living conditions of stakeholders, covering the whole value chain; and (3) sustainability assessment requiring social assessments from the life cycle/organizational perspective [59] | (1) Integration of the improvement of organizational social sustainability conditions and performance (including the whole value chain and all stakeholders) into assessment, development of indicators, and collection of information; (2) life cycle-based social sustainability assessments of organizations/products to improve the living conditions of stakeholders, covering the whole value chain; and (3) social sustainability assessments from the life cycle/organizational perspective |
| Societal LCA of products for the assessment of societal impacts based on many hundreds of specific indicators to support movement towards and monitoring of sustainability and link policy/policymakers to sustainable development [58] | (1) Societal life cycle assessments of products, (2) assessment of societal sustainability impacts, (3) use of many hundreds of specific societal sustainability indicators, and (4) integration of societal aspects of sustainable development into assessment, development of indicators, and collection of information |

| SO-LCA Approaches | Social Sustainability Handprint Assessment Approaches |
|---|---|
| (1) An organizational approach (social impacts on stakeholders based on the conduct of companies engaged in the product life cycle) and (2) human dignity and well-being as an area of protection, including focus on associated positive/negative social impacts [70] | Integration of (1) organizational social sustainability (social sustainability impacts on stakeholders based on the conduct of companies in the full product life cycle) and (2) human dignity and well-being as parts of social sustainability, including positive/negative impacts, into assessment, development of indicators, and collection of information |
| (1) Support of decision-making about sustainability, taking into account the whole value chain, and (2) application of indicators that can be linked to the UN SDGs (to allow policy/industry decision-makers to link performance to each social aspect) [78] | Integration of (1) decision-making about social sustainability (whole value chain), (2) social sustainability indicators that are linked to social/societal aspects of the UN SDGs, (3) performance indicators for all social aspects, and (4) policy/industry decision-making into assessment, development of indicators, and collection of information |

### 3.2.3. S-LCA and SO-LCA Challenges and Limitations

Life cycle approaches comprise many challenges and limitations that need to be considered in the assessment of social sustainability handprints. Previous studies have recognized many challenges and limitations related to S-LCA approaches, encompassing (1) linking of social indicators and impacts to products, lack of assessment of the social performance of products, and product-level data covering whole life cycles [59]; (2) assessment of social benefits and impacts considering the perception of social issues based on various culture, value, and lifestyle-related aspects [65]; and (3) application to company-level assessment of social implications, including identification of specific indicators for detailed analysis of social impacts, the need for more detailed data, data availability (e.g., site-specific data covering all stakeholders and companies involved in the whole life cycle), and general guideline indicators [80].

These findings suggest that the assessment of social sustainability handprints needs to (1) link social sustainability indicators and impacts to products; (2) assess social sustainability performance of products; (3) use product-level social sustainability information/data covering whole life cycles; (4) assess social sustainability benefits and impacts, taking into account various perceptions based on culture, values, and lifestyle-related aspects; (5) assess organizational- and company-level social sustainability implications; (6) identify specific social sustainability indicators; (7) assess social sustainability impacts in detail; and (8) use detailed and site-specific social sustainability information/data covering all stakeholders/companies in the whole life cycle. Further implications of the challenges and limitations associated with S-LCA and SO-LCA approaches for the assessment of social sustainability handprints are presented in Table 11.

**Table 11.** Implications of S-LCA challenges and limitations for the assessment of social sustainability handprints.

| S-LCA Challenges and Limitations | Social Sustainability Handprint Assessment Approaches |
|---|---|
| Improvement of social conditions in the product life cycle due to significant problems related to (1) the creation of an effect through its application and (2) its practical application [84] | Integration of (1) improvement of social sustainability conditions in the product life cycle and (2) creation of positive social sustainability effects through practical application into assessment, development of indicators, and collection of information |
| The operationalization and measurability of social indicators, including limited identification of stakeholder concerns and gathering of data [85] | Integration of (1) operationalization and measurability of social sustainability indicators, (2) comprehensive identification of stakeholder concerns, and (3) the need for high-quality information/data into assessment, development of indicators, and collection of information |

**Table 11.** *Cont.*

| S-LCA Challenges and Limitations | Social Sustainability Handprint Assessment Approaches |
|---|---|
| Access to primary and/or good-quality local, national, and global data that are needed for credible conclusions (e.g., lack of highly integrated and cooperating supply chains and good-quality databases) [86] | Integration of (1) access to primary and good-quality local, national, and global information/data; (2) highly integrated and cooperating supply chains; and (3) good-quality information sources/databases into assessment, development of indicators, and collection of information |
| (1) It is not possible to implement a comprehensive assessment and address all social effects of changes (there are also always unpredictable effects) within product life cycles; (2) quantitative assessment cannot predict some effects and the approach is based on generalization, which creates difficulties because social impacts of changes vary significantly between countries; (3) it will never be able to assess the goodness of specific scenarios (only help rank alternative scenarios within the same overall context); and (4) it cannot predict the absolute improvement in real welfare status as a whole (welfare is more linked to cultural evolution and state action than product chains) [87]. | Integration of (1) comprehensive approaches, including all possible social sustainability effects of changes associated with product life cycles; (2) qualitative approaches, taking into account that impacts of changes vary significantly between countries; (3) goodness of specific scenarios; and (4) absolute improvements in real welfare status as a whole (considering cultural evolution/conditions and state actions) into assessment, development of indicators, and collection of information |
| The assessment of social aspects and analysis of social problems (e.g., access to data and lack of collaboration among certain stakeholder categories) [88] | Assessment of social sustainability aspects and problems based on access to high-quality information/data and collaboration among all stakeholders |
| (1) Collection of good primary social data covering the whole supply chain, including specific processes (e.g., transport) or locations (e.g., primary data from other countries), and (2) limited influence of companies (often SMEs) on decision-making within the supply chain [89] | Integration of (1) collection of good primary social sustainability information/data covering the whole supply chain and specific processes/locations (e.g., other countries) and (2) organizational/company influence on decision-making about social sustainability within the supply chain into assessment, development of indicators, and collection of information |
| (1) The inclusion of cultural values [90], (2) social data quality and availability including process-level data [91], (3) assessment of full life cycles in comparative applications in the company context (companies often prefer assessment tools with a very limited life cycle perspective) [92], and (4) the selection of social indicators (to enable meaningful application) [93] | Integration of (1) cultural values and (2) high-quality information/data (e.g., process level), (3) enhanced access to information/data, (4) full life cycle social sustainability in the company context (comprehensive life cycle perspective), and (5) appropriate selection of social sustainability indicators (to enable meaningful application) into assessment, development of indicators, and collection of information |
| (1) Lack of site-specific data and of awareness of some local stakeholders (e.g., about socio-economic issues and due to low literacy), (2) the selection of appropriate indicators, and (3) availability of consistent data covering various geographic locations [94] | Integration of (1) site-specific information/data, including access to consistent information/data (various geographic locations); (2) awareness/literacy of local stakeholders (e.g., about social/societal sustainability issues); and (3) the selection of appropriate social sustainability indicators into assessment, development of indicators, and collection of information |
| (1) Lack of social data (limits the evaluation of sectoral social performance) [82] and data on, e.g., informal recycling in developing countries [95], and (2) data collection based on reports (corporate sustainability and social responsibility) that do not address social problems within life cycle thinking [96] | Integration of (1) high quality and access to information/data, including informal activities and problems; (2) sectoral social sustainability performance; (3) collection of site-specific information/data in addition to reports; and (4) social sustainability problems based on life cycle thinking into assessment, development of indicators, and collection of information |
| SO-LCA Challenges and Limitations | Social Sustainability Handprint Assessment Approaches |

| S-LCA Challenges and Limitations | Social Sustainability Handprint Assessment Approaches |
|---|---|
| (1) Different situations for different organizations (e.g., life cycle of organizations and data collection/quality), (2) selection and categorization of activities of organizations included in the study (e.g., definition of system boundary), (3) definition of reporting flow for performance tracking, (4) interpretation of broad results, (5) collection of large amounts of primary data from organizations/suppliers, and (6) external/internal communication of results [97,98] | Integration of (1) different social/societal sustainability situations and contexts of organizations (e.g., life cycles and social sustainability information/data collection/quality), (2) comprehensive selection and categorization of assessed organizational social sustainability activities, (3) social sustainability performance monitoring and reporting, (4) interpretation of broad social sustainability results, (5) collection of large amounts of primary social sustainability information/data from organizations/suppliers, and (6) external/internal communication of social sustainability results into assessment, development of indicators, and collection of information |
| (1) Organizational level (e.g., selection of included part); (2) data collection and quality assessment; (3) identification of included activities based on data availability; (4) personnel involvement in data collection and coordination of on-site data collection, company records, and data collection from suppliers; (5) limited assessment of local impacts (using existing tools and methods); (6) large amounts of information and results; (7) relating hotspots to company challenges, easy/readable presentation of data, and time constraints; (8) region-specific databases; and (9) lack of supporting software tools, electronic data collection system, and data on purchased service [99] | Integration of (1) organizational social sustainability level (e.g., selection of parts); (2) social sustainability information/data collection, availability, and quality assessment; (3) identification of included social sustainability activities; (4) personnel involvement in social sustainability information/data collection; (5) coordination of on-site/supplier social sustainability information/data collection; (6) company social sustainability records; (7) local social sustainability impacts; (8) presentation of social sustainability information/results; (9) organizational/company social sustainability hotspots and challenges; (10) region-specific social sustainability information/databases; and (11) social sustainability software tools and electronic social sustainability information/data collection systems/services into assessment, development of indicators, and collection of information |

### 3.2.4. S-LCA Development Focus Areas

There are multiple S-LCA development focus areas that have implications for the assessment of social sustainability handprints. Previous studies have recognized that S-LCA development focus areas encompass, e.g., (1) the identification, development, and selection of social indicators and definition of important assessment focus areas and social impacts [83]; (2) the improvement of indicators [86] and development of indicators for each subcategory [79]; (3) the need for a theoretical basis that is inclusive and flexible and covers a broad range of contexts [85]; (4) consideration of the path of full acknowledgement of existing social science research (implies fundamental questions about methodological foundations and could include, e.g., a review of recent human well-being concepts to inspire a new integrated set of social impact categories) in addition to the current path of copying the LCA approach (which implies, e.g., more research on indicator development) [74]; (5) a set of social criteria taking into account the importance of national level focus (high influence of cultural perceptions on social issues), including links to the international level through combining/comparison [20]; (6) appropriate and sufficient amount of social indicators taking into account whole life cycle perspective (e.g., supply chain), lack of data and the need for improved databases [23], (7) use of both external and internal information sources (e.g., measurement of happiness of employees, production stage studies, approval by suppliers, and consumer satisfaction) [96]; and (8) comprehensive assessment and inventory of all life cycle phases [95].

These findings suggest that the assessment of social sustainability handprints needs to (1) identify, develop, and select social sustainability indicators; (2) define all important (e.g., new and emerging) social sustainability assessment focus areas and impacts; (3) develop participatory approaches; (4) take place within an inclusive, broad, diverse, and flexible framework that takes into account various contexts; (5) integrate both social science (e.g., human well-being) and life cycle-based approaches; (6) consider national and international

social sustainability criteria, including various cultural conditions; (7) apply whole life cycle perspective; and (8) use high-quality information/data, including external and internal sources of information.

Additionally, previous studies have recognized that the development of S-LCA approaches needs to focus on (1) numerous approaches [79] and the inclusion of the social dimension of sustainability thinking [68]; (2) the development of metrics (indicators) and positive contributions (opportunities related to social issues) [82] and improvement of relevance/feasibility in methodological development, taking into account various perspectives) [92]; (3) mapping of many social aspects related to company behavior using qualitative and semi-quantitative indicators [91]; (4) positive and negative impacts covering full life cycles and understanding the improvement opportunities associated with social sustainability considerations of marginalized stakeholder groups (e.g., social impacts on value chain actors and consumers) [85]; (5) exploration of the basic question of what life cycle thinking has to offer for the assessment of social impacts of product chains [84]; (6) a framework that is suitable for the better evaluation of the dynamic, complex, and mutual interactions between social indicators (e.g., to avoid burden shifting) [85]; and (7) the development of participatory approaches and the theoretical framework for the definition of approaches to assess social indicators and impacts [83]. Previous studies have also recognized the following development focus areas:

- Local contextualization of indicators, establishment of stakeholder concerns through participatory approaches, and localization and justification of the relevance of each indicator (indicators cannot be homogenized across all sectors and disciplines) [85];
- Application to (1) real case studies (recognizing that only certain aspects of social sustainability are addressed due to practical/methodological restrictions related to covering the whole life cycle and all associated companies) and (2) companies to promote concrete measures to improve the social performance of the involved companies or to choose companies that behave better than sectoral averages [21];
- Promotion of both sustainability and circularity/sustainable circular economy through (1) holistic approaches and changes in all value chain stages, (2) the identification of socio-economic and environmental hotspots, and (3) priority actions along the whole value chain, including focus on multiple impacts [13];
- Contributions to better decisions, real social impact improvements, and data (e.g., the identification of companies involved in each process) [74], a focus on and social impacts in the product-use phase [68,91], and industry-specific analysis of social aspects and development of indicators based on industry characteristics and inclusion of stakeholders in the evaluation/prioritization of social aspects [91];
- Consideration of the whole life cycle of a product, the availability/comparability of social data (in the context of multiple internationally operating companies), the and use of company-specific data covering complex international supply structures and manufacturing [91].

These findings suggest that the assessment of social sustainability handprints needs to (1) include social sustainability thinking and application of multiple approaches; (2) address positive contributions, improvement opportunities, and other opportunities related to social sustainability; (3) develop approaches that are relevant and feasible, and consider various perspectives, including stakeholder concerns and marginalized groups; (4) be able to address social sustainability performance and cover all social sustainability aspects related to organizations/companies using, e.g., qualitative and semi-quantitative indicators; (5) include all positive and negative social sustainability impacts and industry/company-specific information/data covering full life cycles and international supply chains; (6) apply life cycle thinking to product/supply/value chains; (7) consider dynamic interactions between social sustainability indicators; (8) develop social sustainability indicators that are based on and applicable to local contexts; and (9) contribute to sustainability, informed decision-making, and circular economy. Further implications of S-LCA development focus areas for the assessment of social sustainability handprints are presented in Table 12.

**Table 12.** Implications of S-LCA development focus areas for the assessment of social sustainability handprints.

| S-LCA Development Focus Areas | Social Sustainability Handprint Assessment Approaches |
|---|---|
| (1) Identification of specific indicators for more detailed analysis of social impacts (existing indicators are too general), including more detailed data (obtaining company-level data is already difficult); (2) enhancement of communication between business/supply chain partners (reports on social performance, sharing/provision of data, and development of databases); (3) screening of production processes to determine the main social impacts, improvement opportunities, and risks that may influence the whole supply chain; (4) understanding possible risks and opportunities from the social perspective (in the initial development phase of a production process); and (5) evaluation of the social impact of company performance for internal assessment and optimization purposes [80] | Integration of (1) identified specific social sustainability indicators; (2) detailed analysis of social sustainability impacts; (3) detailed social sustainability information/data (e.g., organizational/company level); (4) communication about social sustainability between business/supply chain partners (e.g., reports, sharing/provision of information/data, and development of databases); (5) the screening of production processes to determine the main social sustainability impacts, improvement opportunities, and risks (whole supply chain); (6) understanding about potential social sustainability risks/opportunities (e.g., in the initial development phase); and (7) social sustainability impacts of company performance into assessment, development of indicators, and collection of information |
| Indices and indicators can be developed based on (1) global, national, and sectoral sustainability standards; (2) stakeholder interviews; and (3) case studies [100]. | Integration of (1) global, national, and sectoral sustainability standards; (2) stakeholder interviews; and (3) case studies into assessment, development of indicators, and collection of information |
| Better identification and assessment of positive impacts [101] and systemic orientation in social performance measurement to cover essential aspects of holistic social performance measurement [102] | Integration of (1) identified positive impacts, (2) holistic social sustainability performance, and (3) systems thinking into assessment, development of indicators, and collection of information |
| Local specificity/dependence (assessment based on site-specific information about local conditions and specific data for companies in the product chain) [103] and case studies covering whole life cycles and investigation of suitable indicators [104] | Integration of (1) local social sustainability specificity/dependence, (2) site-specific information about local conditions, (3) specific data covering all companies in the product chain), (4) case studies (whole life cycles) on social sustainability, and (5) investigation of suitable social sustainability indicators into assessment, development of indicators, and collection of information |
| (1) Involvement and incorporation of social scientists and social science literature, (2) incorporation of values and diverse cultural contexts (e.g., through the application of participatory research and methods to incorporate different worldviews), (3) inclusion of value systems based on social science and other literature, (4) data collection based on integration and standardization with social science methods (e.g., surveys, interviews, and data about how social impacts occur/are felt), and (5) community-anchored techniques for assessing social (and environmental) impacts (e.g., with situation-specific prioritization focus) [105] | Integration of (1) social scientists and social science literature (e.g., focusing on social/societal sustainability), (2) values and diverse cultural contexts, (3) participatory research and methods to incorporate different worldviews, (4) value systems based on social science/other literature, (5) information/data collection based on integration with social science methods (e.g., surveys and interviews and information/data about how social/societal sustainability impacts occur/are felt), and (6) community-anchored techniques for assessing social/societal sustainability impacts (e.g., using situation-specific prioritization focus) into assessment, development of indicators, and collection of information |
| (1) Participatory (e.g., qualitative) approaches to indicators to promote the inclusion of context-specific and ethical issues and (2) a procedure for including qualitative issues such as qualitative indicators [106] | Integration of (1) participatory (e.g., qualitative) approaches to social sustainability indicators, including the inclusion of context specific/ethical issues; and (2) inclusion of qualitative social sustainability issues/indicators into assessment, development of indicators, and collection of information |
| (1) Assessment of social impacts of products, including, e.g., level and applicability of indicators, and (2) assessment that is very dependent on the availability of data [107] | Integration of (1) social sustainability impacts of products, (2) applicability at all levels, and (3) access to/availability/high quality of social sustainability information/data into assessment, development of indicators, and collection of information |
| Participatory approach/stakeholder consultation (including the role of people/actors) to define and select assessment criteria and indicators (particularly because of the specificity of social issues) [108] | Participatory approaches/stakeholder consultation (e.g., people/actors) to define/select social sustainability assessment criteria and indicators (e.g., based on specificity of social sustainability issues) |

Table 12. *Cont.*

| S-LCA Development Focus Areas | Social Sustainability Handprint Assessment Approaches |
|---|---|
| (1) Awareness of the fact that social research paradigms can justify the diversity of approaches and (2) the recognition of the nature of social phenomena (e.g., multiple layers) and of the characteristics of social and management sciences (e.g., multiple paradigms) [109] | Integration of (1) social (sustainability) research, including full diversity of approaches; (2) the nature of social (sustainability) phenomena (e.g., multiple layers); and (3) the characteristics of social (sustainability) and (social sustainability) management sciences (e.g., multiple paradigms) into assessment, development of indicators, and collection of information |
| (1) Consideration of value chain governance in the assessment of social sustainability of products and (2) support from social sciences/other disciplines to identify investigated causal mechanisms, with particular emphasis on the root causes of the main social problems in product chains [110] | Integration of (1) value chain governance (products) and (2) social sciences/other disciplines (e.g., focusing on social sustainability) to identify causal mechanisms related to, and the root causes of, main social sustainability problems in product chains into assessment, development of indicators, and collection of information |
| Assessment criteria that are meaningful and legitimate for stakeholders and that reflect the values of people (e.g., focus on value and context orientation) [111] | Integration of social sustainability assessment criteria (meaningful/legitimate for stakeholders, reflect the values of people, and focus on value/context orientation) into assessment, development of indicators, and collection of information |
| (1) Addressing positive social impacts associated with human interventions, (2) the development of positive indicators that are capable of capturing the contribution of products to the UN SDGs, and (3) indicators for positive impacts in the context of policy impact assessment [112] | Integration of (1) positive social sustainability impacts (human interventions), (2) development of positive social sustainability indicators, (3) the contribution of products to social/societal aspects of the UN SDGs, and (4) social sustainability indicators for positive impacts in the context of policy impact assessment into assessment, development of indicators, and collection of information |
| (1) Strong collaboration within the whole supply chain, including relations between suppliers to promote awareness of social sustainability; (2) improvement of the assessment of multifunctional products; (3) the multidimensionality of a product from a social perspective, considering ecosystem services and associated relations; and (4) evaluation of the social aspects of the ecosystem services, including the development support tools [113] | Integration of (1) social sustainability collaboration in the whole supply chain (e.g., relations between suppliers to promote awareness of social sustainability); (2) multifunctional products; (3) multidimensionality of products from a social sustainability perspective, taking into social/society–environment relationships, and interfaces; and (4) social/societal sustainability aspects of ecosystem services into assessment, development of indicators, and collection of information |
| (1) The integration of participatory approach considering the diversity of stakeholder interests, local knowledge, and meaningful impact categories for stakeholders in various contexts; (2) inclusion of multidisciplinary approaches; and (3) integration of new skills/knowledge [114] | Integration of (1) a participatory approach considering the diversity of stakeholder interests, local knowledge, and meaningful impact categories for stakeholders in various contexts; (2) multidisciplinary approaches; and (3) new skills and knowledge into assessment, development of indicators, and collection of information |
| (1) Application in particular sectors based on the development of ad hoc indicators [115] and (2) local relevance/specificity through the integration of participatory and multicriteria analysis tools and qualitative techniques (e.g., the involvement of local stakeholders and experts) [116] | Integration of (1) applicability in particular sectors based on ad hoc indicators, (2) local relevance/specificity, (3) participatory and multicriteria analysis tools, (4) qualitative techniques, and (5) the involvement of local stakeholders and experts into assessment, development of indicators, and collection of information |
| (1) The development of specific databases; (2) complete, transparent, and openly published sustainability reports by companies and business associations; and (3) participatory approaches to primary information sources (e.g., questionnaires and surveys for company executives and semi-structured individual interviews for representatives of each stakeholder and experts) [117] | Integration of (1) specific databases; (2) complete, transparent, and openly published sustainability reports by companies and business associations; (3) participatory approaches to primary information sources; (4) questionnaires and surveys for company executives; and (5) semi-structured individual interviews for representatives of each stakeholder and experts into assessment, development of indicators, and collection of information |

### 3.2.5. Life Cycle Sustainability Assessment

Life cycle sustainability assessment (LCSA) expands the scope of life cycle thinking to encompass all pillars (environmental, social, and economic) of sustainability and integrates S-LCA, environmental LCA, and life cycle costing (LCC) based on an assessment of environmental, social, and economic issues [49]. LCSA approaches cover all environmental, social, and economic benefits and negative impacts associated with decision-making processes to promote more sustainable products covering their whole life cycles [49,118] and provides multiple benefits for future and potential decision-makers, companies, consumers, and stakeholders [118]. A coherent and practical social indicator approach to products and processes is very challenging due to a vast amount (over 150) of identified social sustainability indicators and the fact that very few indicators can be directly assigned to products and processes (e.g., organizational- and regional-level indicators need to be used, including the establishment of the product relation) [18].

Previous studies have recognized that LCSA approaches (1) can support comprehensive evaluation of sustainability associated with the life cycles of products and/or services [88]; (2) can contribute to (jointly with life cycle thinking) sustainability science that promotes integrated, comprehensive, and participatory approaches [54]; (3) need to evaluate the impacts of systems (human or natural) on areas that need to be protected and maintained over time (e.g., human well-being and ecosystems) and to consider impacts on human well-being (e.g., health and happiness) considering human needs [25]; (4) can address the social dimension of sustainability, including impacts of organizations, products, or processes on society (in addition to measuring the degree of achievement of societal goals and values), including multiple social indicators such as qualitative standards of activities and systems of organizations (e.g., management practices, procedures, and operating principles) [18]; and (5) can address all environmental, social, and economic benefits and negative impacts associated with decision-making processes to promote more sustainable products covering their whole life cycles and be applied by companies, societal organizations, and governments (e.g., contribution to social welfare jointly with a reduction in the use of natural resources and environmental degradation) [49,118].

These findings suggest that LCSA approaches can support the assessment of social sustainability handprints through the integration of (1) all dimensions of sustainability, including a specific focus on social aspects; (2) all positive and negative social impacts; (3) informed decision-making by all types of organizations, companies, and stakeholders; (4) multiple social sustainability indicators, such as local/regional indicators, and qualitative standards of activities and systems of organizations (e.g., management practices, procedures, and operating principles); (5) life cycle thinking and whole life cycle perspective; (6) impacts on human well-being, including social/society-environment relationships and interfaces; (7) impacts of organizations, products, or processes on society and (8) the degree of achievement of societal goals and values; and (9) application by companies, societal organizations, and governments (including contributions to social welfare and sustainability and to more sustainable social/society–environment relationships and interfaces) into assessment, development of indicators, and collection of information. Further implications of LCSA approaches for the assessment of social sustainability handprints are presented in Table 13, and the implications of challenges and limitations associated with LCSA approaches for the assessment of social sustainability handprints are presented in Table 14.

**Table 13.** Implications of LCSA approaches for the assessment of social sustainability handprints.

| LCSA Approaches | Social Sustainability Handprint Assessment Approaches |
|---|---|
| LCSA approaches can be operationalized based on the definition of sustainable development (WCED 1987) [119] and the three pillars of sustainability, including a focus on both social aspects of sustainability and organizational and qualitative approaches [120]. | Integration of (1) social and societal aspects of sustainable development (as defined by the WCED 1987), (2) social pillar and aspects of sustainability, and (3) organizational and qualitative approaches into assessment, development of indicators, and collection of information |
| LCSA is based on the context of sustainability (as defined by the WCED 1987) and is about the assessment of the impacts of product life cycles on (1) meeting the needs of the present generation or the ability of future generations to meet their needs (intra- and intergenerational equity) and (2) poverty in the present generation and maintenance of the stock of capital for the people living in the near or long-term future [121]. | Integration of (1) social and societal aspects of the sustainability context (as defined by the WCED 1987), (2) all impacts of product life cycles, (3) meeting the needs of the present generation, (4) the ability of future generations to meet their needs, (5) intra- and intergenerational equity, (6) poverty in the present generation, and (7) maintenance of the stock of capital for the people living in the near or long-term future into assessment, development of indicators, and collection of information |
| (1) Support of decision-makers to choose sustainable technologies/products and company decision-makers to promote more sustainable production and to design more sustainable products, and (2) provision of a holistic assessment approach for stakeholders, focusing on the implications of the life cycle of a product for society and the environment [49,118] | Integration of (1) decision-making on technologies/products based on social sustainability, (2) social sustainability in product design and production, (3) organizational and company decision-makers, (4) holistic approaches to stakeholders, (5) implications of product life cycles for societies, and (6) social/society-environment relationships and interfaces into assessment, development of indicators, and collection of information |
| Promotion of (1) sustainable development of society (driven by, e.g., a paradigm shift towards sustainability and sustainability performance evaluation methods/tools) and (2) life cycle perspective as inevitable for all sustainability dimensions [18] | Integration of (1) social and societal aspects of sustainable development of societies (e.g., paradigm shift towards social and societal sustainability), (2) social sustainability performance, and (3) life cycle perspective on social and societal dimensions of sustainability into assessment, development of indicators, and collection of information |
| Product sustainability assessment (in the context of sustainability as defined by the WCED 1987) focusing on (1) the extent to which product life cycles affect poverty levels within the current generation and (2) changes in the level of social, human, natural, and produced capital available for the future population [64] | Integration of (1) product social sustainability, (2) the context of social and societal aspects of sustainability (as defined by the WCED 1987), (3) the impacts of product life cycles on poverty within the current generation, and (4) changes in the level of social and human capital available for the future population into assessment, development of indicators, and collection of information |
| Sustainability dimensions are generally understood to encompass (1) measures of welfare (e.g., happiness, aspiration, and need), (2) inter-generational equity (e.g., equity in welfare), (3) intra-generational equity (e.g., equity in welfare among nations and within nations, regions, and local communities), and (4) interspecies equity (welfare of both humans and other living organisms) dimensions [122] | Integration of (1) welfare (e.g., happiness, aspiration, and need), (2) inter-generational equity (e.g., equity in welfare), (3) intra-generational equity (e.g., equity in welfare within nations, regions, and local communities), and (4) interspecies equity (welfare of humans and other living organisms) into assessment, development of indicators, and collection of information |
| Promotion of social learning, mutual feedback, and co-production of knowledge with other stakeholder groups (e.g., society, businesses, and politicians) based on a common process of problem identification and resolution (in the context of sustainability science) [55] | Integration of (1) social learning, (2) mutual feedback, and (3) co-production of knowledge with other stakeholders (e.g., society, politicians, and businesses) based on joint problem identification and resolution (in the context of sustainability science) into assessment, development of indicators, and collection of information |
| (1) A tool to assess sustainability from the life cycle perspective [123] and (2) promotion of sustainability (and sustainable production) and contribution to the implementation of sustainable development goals in multiple sectors [124] | Integration of (1) social and societal sustainability from the life cycle perspective, (2) social aspects of sustainable production, (3) social and societal aspects of sustainable development, and (4) sectoral approaches into assessment, development of indicators, and collection of information |

**Table 14.** Implications of LCSA challenges and limitations for the assessment of social sustainability handprints.

| LCSA Challenges and Limitations | Social Sustainability Handprint Assessment Approaches |
|---|---|
| (1) Sustainability science has not been taken into account in sustainability assessment studies (particularly not in LCSA studies) and is not addressed in LCSA studies, (2) the current form of application cannot be a holistic and transdisciplinary framework for sustainability, and (3) the limitations of life cycle methods indicate the need for complementary application of multiple approaches [24]. | Integration of (1) sustainability science (including in the LCSA context), (2) holistic approaches, (3) transdisciplinary framework, and (4) multiple and complementary approaches (in addition to life cycle-based approaches) into assessment, development of indicators, and collection of information |
| Approaching sustainability by balancing environmental, social, and economic dimensions (e.g., neglect of environmental carrying capacity may compromise the meeting of human needs linked to social and economic sustainability in the long-term) [122] | Integration of (1) social/society-environment relationships and interfaces, (2) long-term social sustainability, (3) meeting of human needs, and (4) environmental carrying capacity into assessment, development of indicators, and collection of information |
| Social aspects are less addressed and there are practical difficulties related to indicators and data [125] | Integration of (1) social sustainability aspects and (2) practical social sustainability indicators and information/data into assessment, development of indicators, and collection of information |
| (1) Assessment of social impacts on multiple stakeholders (e.g., value chain actors and consumers) and (2) overall upstream and downstream consequences of organizational conduct (indicates lack of life cycle thinking, including the big picture of social performance in life cycles and supply chains) [126] | Integration of (1) social sustainability impacts on multiple stakeholders (e.g., value chain actors and consumers), (2) overall upstream and downstream social sustainability consequences of organizational conduct, and (3) life cycle thinking, including the big picture of social sustainability performance in life cycles and supply chains, into assessment, development of indicators, and collection of information |
| Lack of holistic point of view, life cycle perspective, and focus on multiple environmental impacts in current sustainability assessment tools [127] | Integration of (1) a holistic point of view, (2) life cycle perspective, and (3) multiple social sustainability impacts into assessment, development of indicators, and collection of information |
| (1) Limited to hotspot analysis of systems, products, or services (assessment of direct impacts), and (2) indirect impacts (consequences) are not addressed [128] | Integration of indirect and direct social sustainability impacts and consequences (e.g., related to systems, products, and services) into assessment, development of indicators, and collection of information |

### 3.2.6. LCSA Development Focus Areas

Previous studies have recognized many LCSA development focus areas that have implications for the assessment of social sustainability handprints, including (1) the definition of sustainable development (as defined by the WCED 1987) in the context of LCSA and the way to capture impacts on social and natural capital and on surroundings of product life cycles (e.g., income gains for the poor) [64]; (2) the inclusion of culture through participatory research approaches (e.g., gathering of community-based information) and better understanding of life cycle effects on cultural aspirations [90]; (3) development of life cycle methods towards the proactive enhancement of positive impacts (broader focus than negative impacts) in a way that contributes to sustainable development, the development of positive solutions, and social learning and adaptation [125]; (4) sustainability-oriented holistic approaches and assessment perspectives [54,55]; (5) system-wide approaches and multi-scale (geographical and temporal) perspectives [54]; and (6) socially-embedded and transparent assessment frameworks, including the integration of life cycle and other methods [24]. In addition, previous studies have recognized the following relevant development focus areas:

- Better involvement and participation of stakeholders and a shift from multidisciplinarity towards transdisciplinarity (as in all sustainability assessments) [54];
- Life cycle methods that are embedded in normative, systemic, strategic, and transdisciplinary research frameworks (transparently include multiple approaches, com-

petencies, and perspectives from diverse actors) and complementary application of multiple approaches [24];

- S-LCA development, including social assessment (e.g., stakeholder perspectives), development of guidelines to promote the applicability of social indicators, and new agreement and consensus on unclear social goals and targets in the international or regional context [128];
- Wider integrated assessment goals, including (1) proactive enhancement of positive impacts that contribute to sustainable development, (2) incorporation of sustainability goals, (3) moving towards much broader solution-oriented approach and scope, (4) tailoring the assessment for local and specific impacts (environmental, social, or economic), and (5) interaction among stakeholders (e.g., scientific community, business associations, and policymakers covering many levels, values, visions, and data provision) [55];
- Development of S-LCA approaches, including (1) positive promotion of sustainability through the development of more positive assessment criteria (benefits) and indicators (areas of promotion), (2) the definition of what is good or bad in certain indicators, (3) the incorporation of social and environmental consequences of processes, and (5) the presence of both benefits and impacts [128].

These findings suggest that the assessment of social sustainability handprints needs to integrate (1) social and societal aspects of sustainable development (e.g., as defined by the UN SDGs and the WCED 1987); (2) all impacts of product/service life cycles, including impacts of surrounding societies, activities, and actors; (3) culture, cultural aspirations, community-based information, and involvement of and interaction among diverse stakeholders and actors (e.g., researchers, all types of organizations, and policymakers); (4) participatory, transdisciplinary, transparent, holistic, and system-wide approaches; (5) social learning and adaptation; (6) positive impacts and solutions and local/specific impacts; (7) normative, transdisciplinary, socially-embedded, and systemic life cycle approaches; (8) international, regional, and local social goals and targets; (9) positive impacts that contribute to sustainable development and sustainability goals; and (10) positive criteria (benefits) and indicators (area of promotion), including normative definition of what is good or bad and social and environmental consequences of various activities into assessment, development of indicators, and collection of information. Further implications of development focus areas associated with LCSA approaches for the assessment of social sustainability handprints are presented in Table 15.

**Table 15.** Implications of LCSA development focus areas for the assessment of social sustainability handprints.

| LCSA Development Focus Areas | Social Sustainability Handprint Assessment Approaches |
|---|---|
| (1) Development of life-cycle-based approaches to promote the achievement of sustainability goals and (2) proactive enhancement of positive impacts (beyond comparison of alternatives and avoidance of negative impacts) [54] | Integration of (1) life-cycle-based approaches, (2) the achievement of social sustainability goals, and (3) proactive enhancement of positive social sustainability impacts into assessment, development of indicators, and collection of information |
| Sustainability assessments need to be (1) systemic (e.g., scope, scales, and environmental, social, and economic interrelations/impacts), (2) normative (e.g., incorporation of sustainability principles and dimensions and context-specific perceptions), (3) strategic (e.g., purpose, decision, and action support; broader context; and consideration of alternatives), and (4) transdisciplinary (e.g., knowledge, actors, and stakeholder engagement) [24]. | Integration of (1) systemic (e.g., scope, scales, and social/societal-environment relationships and interfaces), (2) normative (e.g., social sustainability principles, dimensions, and context-specific perceptions), (3) strategic (e.g., broader context, consideration of alternatives and purpose, and decision and action support), and (4) transdisciplinary (e.g., knowledge, actors, and stakeholder engagement) approaches into assessment, development of indicators, and collection of information |
| (1) Selection of indicators based on the UN SDGs and (2) goal-based indicator set encompassing all sustainability dimensions [129] | Integration of (1) social and societal sustainability indicators based on the UN SDGs and (2) goal-based social sustainability indicator sets encompassing social and societal sustainability dimensions into assessment, development of indicators, and collection of information |

**Table 15.** *Cont.*

| LCSA Development Focus Areas | Social Sustainability Handprint Assessment Approaches |
|---|---|
| (1) Culturally inclusive process and additional cultural indicators (and/or dimensions of existing indicators that represent cultural values) [130] and (2) representation of culture and addressing cultural values to present information about social and cultural aspects, diverse cultures, and values to decision-makers (simultaneous consideration of a wide range of impacts) to enhance stakeholder understanding and acceptance of results [130] | Integration of (1) culture and diverse cultures, (2) cultural indicators, (3) cultural values, (4) social and cultural aspects, (5) informed decision-making, (6) a wide range of social and cultural impacts, and (7) the enhancement of stakeholder understanding into assessment, development of indicators, and collection of information |
| Progress towards sustainability based on better integrated assessment approaches and mainstreaming of life cycle thinking to support strategic policy in addition to product development [55] | Integration of (1) progress towards social sustainability, (2) integrated approaches, and (3) mainstreaming of life cycle thinking to support strategic policies and product development into assessment, development of indicators, and collection of information |
| (1) Provision of a set of indicators based on more experience from the application of the approach, (2) more guidance on stakeholder involvement, and (3) ways to address the perspective of future generations [131] | Integration of (1) sets of social sustainability indicators, (2) application experiences, (3) guidance on stakeholder involvement, and (4) perspectives of future generations into assessment, development of indicators, and collection of information |
| (1) More holistic understanding of sustainability, including consideration of both the interactions of the three pillars of sustainability and multi-scale (geographical/temporal) perspectives [123], and (2) the definition of sustainability dimensions [132] | Integration of (1) holistic understanding of social and societal sustainability, including social/societal-environment relationships and interfaces; (2) multi-scale (geographical and temporal) perspective; and (3) the definition of social and societal sustainability into assessment, development of indicators, and collection of information |
| (1) Representation of culture to address cultural needs and concerns of end-users and (2) a participatory approach including continuous communication to promote acceptance of results, understanding of the process, and better control and access to information by the participants [90] | Integration of (1) culture, (2) cultural needs and concerns of end-users, (3) participatory approaches, (4) continuous communication, and (5) better access to and control of information by the participants into assessment, development of indicators, and collection of information |

## 4. Conclusions

The findings of this study suggest that social sustainability handprints can be created through multiple ways, such as positive changes, actions, innovations, and impacts, and by various actors, such as all types of organizations, companies, societal groups, actors, and individuals. In addition, they can be created at many levels, such as local, organizational, company, product/service, process, regional, national, and international. Similarly, social sustainability handprint assessments can apply multiple approaches, such as handprint and life cycle thinking and approaches (e.g., S-LCA and LCSA), sustainability management, assessment and indicators, and sustainability science and focus on many levels. The creation and assessment of social sustainability handprints can take place within various contexts, locations, cultures, and temporal/geographical scales considering specific conditions, characteristics, and perspectives.

In general, the creation and assessment of social sustainability handprints should be linked to the overall frameworks of sustainable development and sustainability in addition to numerous specific ways to create and assess them. The creation of social sustainability handprints can be used to improve the normal level of social sustainability performance associated with organizations, companies, society, societal actors, a group of people, individuals, products, services, processes, or activities. Multiple assessment approaches can be applied to the assessment of both the normal and the improved (as a result of the creation of one or more social sustainability handprints) level of social sustainability performance, including approaches based on sustainability science and research, sustainability management, assessment and indicators, and handprint and life cycle thinking and approaches.

Social sustainability handprints support overall social sustainability management because they require in-dept understanding of and knowledge about social and societal sustainability, including social/society-environment relationships and interfaces. Additionally, they require proactive management and assessment and continuous learning. The findings provide multiple approaches that can be applied in, for example, all types or organizations, as well as many challenges, limitations, and development focus areas that can be addressed as a part of organizational social sustainability management and assessment, including specific focus on the development of social sustainability handprint approaches. Future research should focus on both theoretical and practical aspects of the creation and assessment of social sustainability handprints, including transdisciplinary approaches and real-life case studies covering all types of organizations in both private and public sectors. Focus is needed on all aspects, such as ways to create handprints and assessment approaches, the development of new indicators, high-quality information/data (e.g., local and site-specific), participatory approaches, and social/society–environment relationships and interfaces. In addition, further development of social sustainability handprints should take place within all relevant study contexts, such as sustainability science, sustainability management, assessment and indicators and handprint and life cycle thinking and approaches.

**Funding:** This research was funded by the Kone Foundation, 202006340.

**Acknowledgments:** This study was supported by the Kone Foundation and their support made it possible to carry out this study.

**Conflicts of Interest:** Authors declare no conflict of interest.

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
