# Peer review of "Exploring Social Sustainability Handprint—Part 1: Handprint and Life Cycle Thinking and Approaches"

_sustainability, doi:10.3390/su132011286_

Round 1

Reviewer 1 Report

The authors explored social sustainability handprint with a specific emphasis on the handprint and life cycle thinking and approaches.

  1. The authors need to highlight the novelty and contributions.
  2. The authors claimed that their findings suggest that social sustainability handprints can be used to promote, manage and assess actions, changes and impacts to promote social sustainability including social/society-environment relationships and interfaces. Please, discuss this more in detail.
  3. Please, incorporate conclusion into the manuscript. 
  4. Overall, the manuscript is well written. I recommend it for a minor revision

Author Response

Reviewer 1

The authors explored social sustainability handprint with a specific emphasis on the handprint and life cycle thinking and approaches.

  1. The authors need to highlight the novelty and contributions.

Response: The chosen approach is creative, innovative and highlights novelty because there are no similar studies on social sustainability handprints. (this sentence was also added to the materials/methods section). In addition, this was added to the abstract: …It addressed a clear gap in research…

2. The authors claimed that their findings suggest that social sustainability handprints can be used to promote, manage and assess actions, changes and impacts to promote social sustainability including social/society-environment relationships and interfaces. Please, discuss this more in detail.

Response: Please note that a major revision has taken place and the focus is much more clear now.  The focus is on the creation and assessment of social sustainability handprints. The findings follow this approach and suggest multiple ways to create and multiple approaches to assess social sustainability handprints including issues that should be integrated into assessments. Social sustainability handprints as an overall approach/framework can be used to manage and assess these things (dynamic interplay between normal practice and continuous improvements including continuous management for changes and actions and creation of impacts supported by continuous assessment of performance/achievements ect).

Social/society-environment relationships and interfaces are an important part of the approach (therefore mentioned) but not assessed in detail in this study eventhough some aspects are addressed in the context of environmental, ecological and carbon handprints. In addition, these aspects are addressed in many of the referenced key elements and approaches. Future studies are needed and this is noted. These issues are also discussed both in the results and discussion and the conclusions section.

3. Please, incorporate conclusion into the manuscript. 

Response: Conclusions section was added.

4. Overall, the manuscript is well written. I recommend it for a minor revision

Response: Thank you for your positive comment and after this major revision it should even better.

Reviewer 2 Report

The research is innovative, but there are still some problems:

1. The article pays more attention to the cross-sectional research of handprint, carbon handprint, social sustainability handprint, etc., but the longitudinal depth of the research is insufficient.

2. The paper used too many tables to introduce the basic connotation of various types of handprints, which leads to confusion in the overall logic of the article and too little content for in-depth analysis.

3. Why are the titles of many tables the same? e.g. 1-2,3-4,6-7,8-9,10-12,16-21,22-23,24-28,29-30 and 34-35.

4. The content of the paper is relatively loose, and the overall logic of the paper needs to be improved. It is recommended not to pay attention to too many contents, and focus on about 3 contents for in-depth analysis.

5. The theoretical contribution and practical enlightenment need to be clearly given in the article.

Author Response

Reviewer 2

The research is innovative, but there are still some problems:

Response: Thank you for this comment and I tried to explore this topic in a way that contributes to further development/research by me and hopefully others. I have carried out a major revision as suggested.

1. The article pays more attention to the cross-sectional research of handprint, carbon handprint, social sustainability handprint, etc., but the longitudinal depth of the research is insufficient.

Response: A major revision has been implemented and the approach is much more focused and in-depth now.

2. The paper used too many tables to introduce the basic connotation of various types of handprints, which leads to confusion in the overall logic of the article and too little content for in-depth analysis.

Response: A  major revision has been carried out and the overall logic is much better now including in-depth analysis.

3. Why are the titles of many tables the same? e.g. 1-2,3-4,6-7,8-9,10-12,16-21,22-23,24-28,29-30 and 34-35.

Response: A major revision has taken place and there are now only 14 tables (none of which has the same name).

4. The content of the paper is relatively loose, and the overall logic of the paper needs to be improved. It is recommended not to pay attention to too many contents, and focus on about 3 contents for in-depth analysis.

Response: Thank you for this constructive comment and I have implemented a major revision. I kept the overall topics since they are relevant and deserve attention but focused on 2 contents (creation and assessment) in my in-depth analysis.

5. The theoretical contribution and practical enlightenment need to be clearly given in the article.

Response: A major revision has been implemented and the new focus on the creation and assessment of social sustainability handprints is applicable to, for example, all types of organizations which can integrate the approach to their management and assessment. Future studies will address real-life case studies and this study was an initial exploration of the topic. The theoretical framework provided in the materials and methods add to theory building in this field in addition to findings and the simplified approach presented in that section is a novel approach that expands and builds on existing knowledge

Round 2

Reviewer 2 Report

Considering that the author addressed the main concerns satisfactory, I have no more comments.